# On the Generalizability and Predictability of Recommender Systems

**Duncan McElfresh**[*1], **Sujay Khandagale**[*1], **Jonathan Valverde**[*1,3],
**John P. Dickerson**[2,3], **Colin White**[1]

[1]Abacus.AI, [2]ArthurAI, [3]University of Maryland

## Abstract

While other areas of machine learning have seen more and more automation, designing a high-performing recommender system still requires a high level of human effort. Furthermore, recent work has shown that modern recommender system algorithms do not always improve over well-tuned baselines. A natural follow-up question is, "how do we choose the right algorithm for a new dataset and performance metric?" In this work, we start by giving the first large-scale study of recommender system approaches by comparing 24 algorithms and 100 sets of hyperparameters across 85 datasets and 315 metrics. We find that the best algorithms and hyperparameters are highly dependent on the dataset and performance metric. However, there is also a strong correlation between the performance of each algorithm and various meta-features of the datasets. Motivated by these findings, we create RecZilla, a meta-learning approach to recommender systems that uses a model to predict the best algorithm and hyperparameters for new, unseen datasets. By using far more meta-training data than prior work, RecZilla is able to substantially reduce the level of human involvement when faced with a new recommender system application. We not only release our code and pretrained RecZilla models, but also all of our raw experimental results, so that practitioners can train a RecZilla model for their desired performance metric: https://github.com/naszilla/reczilla.

## 1 Introduction

Due to the large computational resources for training machine learning models, researchers have found many ways to repurpose existing computation. For example, it is common to start with a pretrained ImageNet classification model for computer vision tasks [30, 50, 75], or a pretrained BERT model for natural language tasks [31, 65]. These approaches work well because the core problems are largely homogeneous; for example, any computer vision model at its core must be able to distinguish edges, colors, and shapes. Even a task-specific architecture can be found automatically through neural architecture search [36], since the building blocks such as convolutional layers stay the same.

On the other hand, recommender system (rec-sys) research has followed a different trajectory: despite their widespread usage across many e-commerce, social media, and entertainment companies such as Amazon, YouTube, and Netflix [21, 41, 77], there is far less work in reusing models or automating the process of selecting models. Many rec-sys techniques are designed and optimized with just a *single* dataset in mind [21, 41, 48, 58, 81]. Intuitively, this might be because each rec-sys application is highly unique based on the dataset and the target metric. For example, a typical user session looks very different among e.g. YouTube, Home Depot, and AirBnB [21, 48, 58]. However, this intuition

---

[*]Equal Contribution. Work done while the first two authors were employed at Abacus.AI. Correspondence to: {duncan, sujay, colin}@abacus.ai, valverde@cs.umd.edu, john@arthur.ai

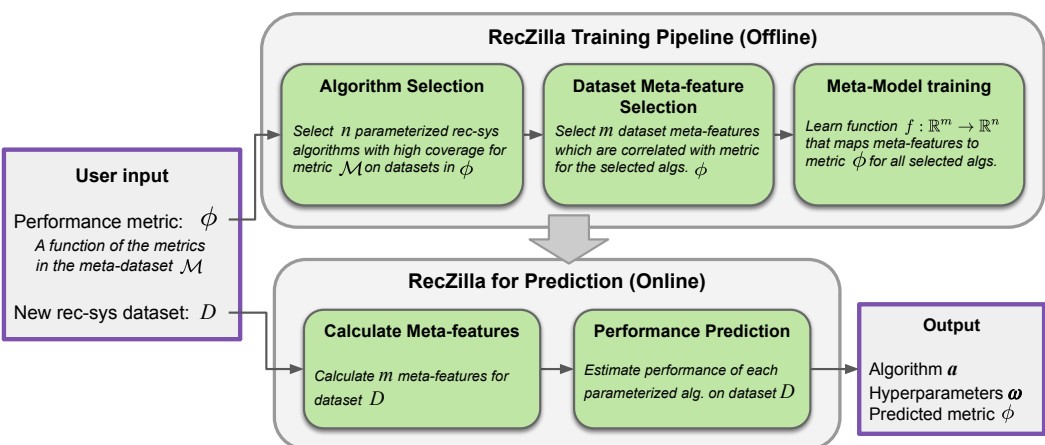

Figure 1: RecZilla recommends a parameterized rec-sys algorithm for a user-provided dataset and performance metric. The RecZilla pipeline is built using a meta-dataset $\mathcal{M}$ that includes many different performance metrics evaluated on many different rec-sys algorihtms on many different datasets; we estimate algorithm performance using dataset meta-features. To avoid over-fitting and reduce runtime, users can limit the number of algorithms and meta-features to constants $n$ and $m$.

has not been formally established. Furthermore, recent work has shown that neural recommender system algorithms do not always improve over well-tuned baselines such as $k$-nearest neighbor and matrix factorization [28]. A natural question is then, "how do we choose the right algorithm for a new dataset and performance metric?"

In this work, we show that the best algorithm and hyperparameters are highly dependent on the dataset and user-defined performance metric. Specifically, we run the first large-scale study of rec-sys approaches by comparing 24 algorithms across 85 datasets and 315 metrics. For each dataset and algorithm pair, we test up to 100 hyperparameters (given a 10 hour time limit per pair). The codebase that we release, which includes a unified API for a large, diverse set of algorithms, datasets, and metrics, may be of independent interest. We show that the algorithms do not *generalize* – the set of algorithms which perform well changes substantially across dataset and across performance metrics. Furthermore, the best hyperparameters of a rec-sys algorithm on one dataset often perform significantly worse than the best hyperparameters on a different dataset. Although we show that there are no universal algorithms that work well on most datasets, we *do* show that various meta-features of the dataset can be used to *predict* the performance of rec-sys algorithms. In fact, the same meta-features are also predictive of the runtime of rec-sys algorithms as well as the "dataset hardness" – how challenging it is to find a high-performing model on a particular dataset.

Motivated by these findings, we introduce RecZilla, a meta-learning-based algorithm selection approach (see Figure 1) inspired by SATzilla [83]. At the core of RecZilla is a model that, given a user-defined performance metric, predicts the best rec-sys algorithm and hyperparameters for a new dataset based on meta dataset features such as number of users and items, and spectral properties of the interaction matrix. We show that RecZilla quickly finds high-performing algorithms on datasets it has never seen before. While there has been prior work on meta-learning for recommender systems [26, 27], no prior work is metric-independent, searches for hyperparameters as well as algorithms, or considers more than nine families of datasets. By running an ablation study on the number of meta-training datasets, we show that more datasets are crucial to the success of RecZilla. We release ready-to-use, pretrained RecZilla models for common test metrics, and we release the raw results from our large-scale study, along with code so that practitioners can easily train a new RecZilla model for their specific performance metric of interest.

**Our contributions.** We summarize our main contributions below.

- We run a large-scale study of recommender systems, showing that the best algorithm and hyper-parameters are highly dependent on the dataset and user-defined performance metric. We also show that dataset meta-features are predictive of the performance of algorithms.

- We create RecZilla, a meta-learning-based algorithm selection approach which, given a performance metric, efficiently predicts the best algorithm and set of hyperparameters on new datasets.
- We release a public repository containing 85 datasets and 24 rec-sys algorithms, accessed through a unified API. Furthermore, we release both pretrained RecZilla models, and raw data so that users can train a new RecZilla model on their desired metric.

**Related Work**    Recommender systems are a widely studied area of research [10]. Common approaches include $k$-nearest neighbors [1], matrix factorization [57, 63], and deep learning approaches [21, 41, 77]. For a survey on recommender systems, see [4, 10]. A recent meta-study showed that of the 12 published neural rec-sys approaches published at top conferences between 2015 and 2018, 11 performed worse than well-tuned baselines (e.g. nearest neighbor search or linear models) [28]. Another recent paper found that the relative performance of rec-sys algorithms can change significantly based on the choice of datasets used [14].

Algorithm selection for recommender systems was first studied in 2011 [52] by using a graph representation of item ratings. Follow-up work used dataset meta-features to select the best nearest neighbor and matrix factorization algorithms [3, 35, 43]. Subsequent work focused on improving the model and framework [27] including studying 74 meta-features systematically [23]. More recent approaches from 2018 run meta-learning for recommender systems by casting the meta-problem itself as a collaborative filtering problem. Performance is then estimated with subsampling landmarkers [24, 25, 26]. No prior work in algorithm selection for rec-sys includes open-source Python code. There is also work on automated machine learning (AutoML) for recommender systems, without meta-learning [6, 46, 47, 82]. Finally, we note that meta-learning across rec-sys datasets is not to be confused with the body of work on meta-learning user preferences *within a single* rec-sys dataset [17, 18, 19]. To the best of our knowledge, no meta-learning or AutoML rec-sys paper has run experiments on more than nine dataset families or four test metrics, and no prior work predicts hyperparameters in addition to algorithms.

## 2    Analysis of Recommender Systems

In this section, we present a large-scale empirical study of rec-sys algorithms across a large, diverse set of datasets and metrics. We assess the following two research questions.

1. **Generalizability.** If a rec-sys algorithm or set of hyperparameters performs well on one dataset and metric, will it perform well on other datasets or on other metrics?
2. **Predictability.** Given a metric, can various dataset meta-features be used to predict the performance of rec-sys algorithms?

**Algorithms, datasets, and metrics implemented.**    We present full results for 20 rec-sys algorithms, including methods from recent literature and common baselines. Methods include five similarity and clustering-based methods: User-KNN [74], Item-KNN [76], P3Alpha [20], RP3Beta [72], and Co-Clustering [39]; six Matrix-Factorization (MF) methods: MF-FunkSVD, MF-AsySVD [56], MF-BPR [73], IALS [51], Pure-SVD, Non-negative matrix factorization (NMF) [22]; five methods based on linear models: Global-Effects, SLIM-BPR [7], SLIM-Elastic-Net [62], EASE-R [78], and SlopeOne [61]; two simple baselines: Random and Top-Pop; and two neural network based methods: User-NeuRec [84] and Mult-VAE [64]. We also include partial results (on 8-10 datasets each) for four more neural network based methods: DELF-EF [13], DELF-MLP [13], Item-NeuRec [84], and Spectral-CF [85] (with results included in Tables 9, 10, and 11). These algorithms were chosen due to their high performance, popularity, and speed. For many algorithms, we used the implementations from the codebase of Dacrema et al. [28]. For full details of the algorithms, see Appendix A.

We run the algorithms on 85 datasets from 19 dataset "families": Amazon [71], Anime [16], BookCrossing [87], CiaoDVD [45, 59], Dating (Libimseti.cz) [59, 60], Epinions [67, 68], FilmTrust [44], Frappe [8], Gowalla [15], Jester2 [40], LastFM [11], MarketBias-Electronics and MarketBias-ModCloth [80], MovieTweetings [33], Movielens [49], NetflixPrize [9], Recipes [66], Wikilens [37], and Yahoo [34]. Here, a "dataset family" refers to an original dataset, while "dataset" refers to a single train-test split drawn from the original dataset, which may be a small subset of the original. We implemented the majority of rec-sys datasets commonly used for research; to the best of our knowledge, this is the largest number of rec-sys datasets accessible in a single open-source repository.

Table 1: The relative performance of each rec-sys algorithm depends on the dataset and metric. This table shows the mean, min (best) and max (worst) rank achieved by all 20 algorithms over all 85 datasets, over 10 accuracy and hit-rate metrics at all cutoffs tested. This includes metrics NDCG, precision, recall, Prec.-Rec.-Min-density, hit-rate, F1, MAP, MAP-Min-density, ARHR, and MRR.

| Rank | Item-KNN | P3alpha | SLIM-BPR | EASE-R | RP3beta | SVD | SLIM-ElasticNet | iALS | NMF | User-KNN | MF-Funk | TopPop | MF-Asy | MF-BPR | Mult-VAE | U-neural | GlobalEffects | CoClustering | Random | SlopeOne |
|---|---|---|---|---|---|---|---|---|---|---|---|---|---|---|---|---|---|---|---|---|
| Min. | 1 | 1 | 1 | 1 | 1 | 1 | 1 | 1 | 1 | 1 | 1 | 1 | 1 | 1 | 1 | 1 | 2 | 1 | 9 | 7 |
| Max. | 14 | 18 | 14 | 18 | 17 | 16 | 17 | 19 | 14 | 17 | 18 | 19 | 16 | 17 | 20 | 20 | 20 | 19 | 20 | 20 |
| Mean | 2.3 | 4.2 | 4.7 | 5.3 | 6 | 6 | 7 | 7 | 7.1 | 7.6 | 9.4 | 10.4 | 10.7 | 11.2 | 11.7 | 12.3 | 13.3 | 14.9 | 16.2 | 16.7 |

We use 23 different "base" metrics: ARHR, Average Popularity, Diversity Similarity, F1 Score, Gini Index, Herfindahl Index, Hit Rate, Item Coverage, Item-hit Coverage, Precision (PREC), Precision-Recall Min Denominator, Recall, MAP, MAP Min Denominator, Mean Inter-List Diversity, MRR, NDCG, Novelty, Shannon Entropy, User Coverage, User-hit Coverage. All of these metrics are computed at cutoffs $\{1, 2, 3, 4, 5, 6, 7, 8, 9, 10, 15, 20, 30, 40, 50\}$, for a total of 315 different metrics. See Appendix A.6 for more details. These metrics include the most popular from the literature, and we use the Dacrema et al. [28] implementations.

**Experimental design.** Each dataset's train, validation, and test split is based on leave-last-$k$-out (and our repository also includes splits based on global timestamp). We use a random hyperparameter search for all methods, with the exception of neural network based methods. Since neural networks require far more resources to train (longer training time, and requiring GPUs), we use only the default hyperparameters for neural algorithms. For each non-neural algorithm, we expose several hyperparameters and give ranges based on common values. For each dataset, we run each algorithm on a random sample of up to 100 hyperparameter sets. Each algorithm is allocated a 10 hour limit for each dataset split; we train and test the algorithm with at most 100 hyperparameter sets on an `n1-highmem-2` Google Cloud instance, until the time limit is reached. Each neural network method is trained on each dataset using the default hyperparameters used in its respective paper, with a time limit of 15 hours on an NVIDIA Tesla T4 GPU. All neural network methods are trained with batch size 64, for up to 100 epochs; early stopping occurs if loss does not improve in 5 epochs.

Each algorithm is trained on the train split, and the performance metrics are computed on the test split. We refer to each combination of (algorithm, hyperparameter set, dataset) as an *experiment*. By running 24 algorithms, most with up to 100 hyperparameters, on 85 datasets, this resulted in 84 850 successful experiments, and by computing 315 metrics, our final meta-dataset of results includes more than 26 million evaluations. We give a more detailed look at the breakdown of experiments in Appendix A, and we discuss any potential biases in the resulting dataset in Section 4.

## 2.1 On the generalizability of rec-sys algorithms

*If a rec-sys algorithm or set of hyperparameters performs well on one dataset and metric, will it perform well on other datasets or on other metrics?*

Our first observation is that *all* algorithms perform *well* on some datasets, and poorly on others. First we identify the best-performing hyperparameter set for each (algorithm, dataset) pair—to simulate hyperparameter optimization using our meta-dataset. We then rank all algorithms for each dataset, according to several performance metrics.

If we focus on a single metric, then many algorithms are ranked first according to this metric on at least one dataset. Take for example metric NDCG@1: 17 of 20 algorithms are ranked either first or second on at least one dataset. The same is true for metric RECALL@50: all algorithms except for SlopeOne, GlobalEffects, and Random are ranked either first or second on at least one dataset. The same is true for many other metrics and cutoffs (see Table 9 in Appendix C).

Average performance is more varied: some algorithms tend to perform better than others. Table 1 shows the mean, min (best) and max (worst) ranking of all 24 algorithms over all dataset and all

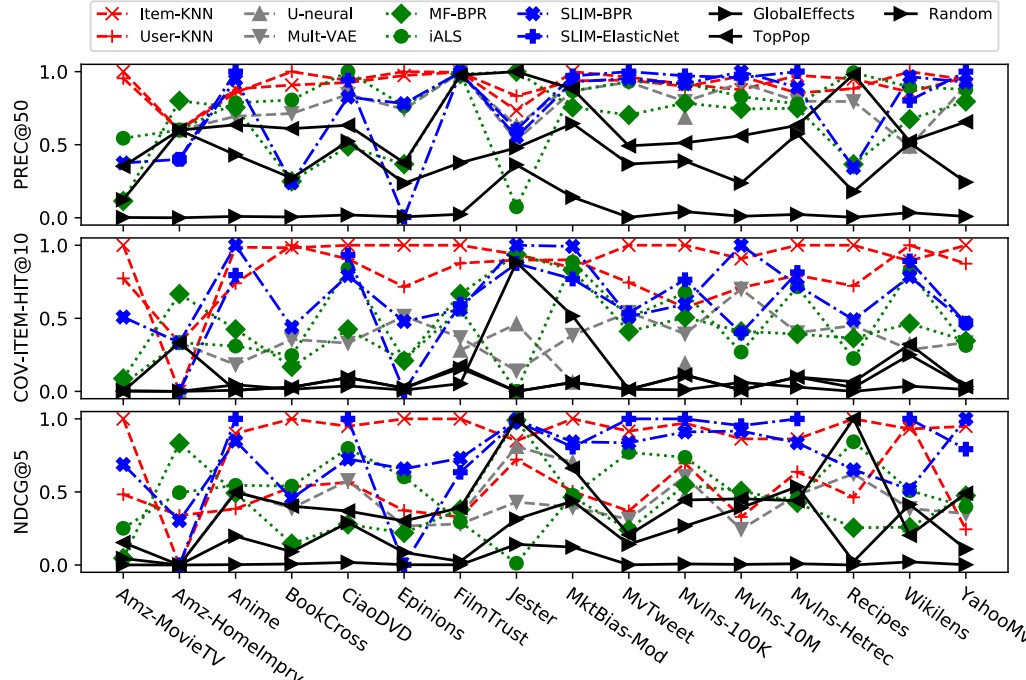

Figure 2: Relative algorithm performance depends on both the dataset and metric: no algorithm dominates across all dataests or metrics. Each plot shows a different metric, normalized to $[0, 1]$ for each dataset; the horizontal axis shows different dataset, ordered alphabetically. Each series corresponds to a different algorithm: similarity-based methods are red, matrix factorization methods are green, baseline methods are black, and neural network methods are in gray.

accuracy and hit-rate metrics. This includes metrics NDCG, precision, recall, Prec.-Rec.-Min-density, hit-rate, F1, MAP, MAP-Min-density, ARHR, and MRR (see Appendix A.6 for descriptions of these metrics). Nearly all algorithms are ranked first for at least one metric on at least one dataset. Many algorithms perform well on average. Furthermore, most algorithms perform very poorly in some cases: the maximum rank is at least 14 (out of 20) for all algorithms.

To illustrate the changes in algorithm performance across datasets, Figure 2 shows the normalized metric values for eight algorithms across 17 dataset splits. Some algorithms tend to perform well (Item-KNN and SLIM-BPR) and others poorly (Random, TopPop), but no algorithm clearly dominates for all metrics and datasets. This is a primary motivation for our meta-learning pipeline descirbed in Section 3: different algorithms perform well for different datasets on different metrics, so it is important to identify appropriate algorithms for each setting.

**Generalizability of hyperparameters.** While the previous section assessed the generalizability of pairs of (algorithm, hyperparameters), now we assess the generalizability of the hyperparameters themselves while keeping the algorithms fixed. For a given rec-sys algorithm, we can tune it on a dataset $i$, and then evaluate the normalized performance of the tuned method on a dataset $j$, compared to the normalized performance of the best hyperparameters from dataset $j$. In other words, we compute the performance of tuning a method on one dataset and deploying it on another.

In Figure 3, we run this experiment for all pairs of datasets (one dataset per dataset family). We plot the hyperparameter transfer for three different algorithms, as well as the average over all algorithms which completed sufficiently many experiments across the set of hyperparameters. For each given $i$, $j$, we create the set of hyperparameters that completed for the given algorithm on both datasets $i$ and $j$, and then we use min-max scaling for the performance metric values of these hyperparameters on $i$ and on $j$ separately. Therefore, all matrix values are between 0 and 1; a value close to 1 indicates that the best hyperparameters from dataset $i$ are also nearly the best on dataset $j$. A value close to 0 indicates that the best hyperparameters from dataset $i$ are nearly the worst for dataset $j$. Across

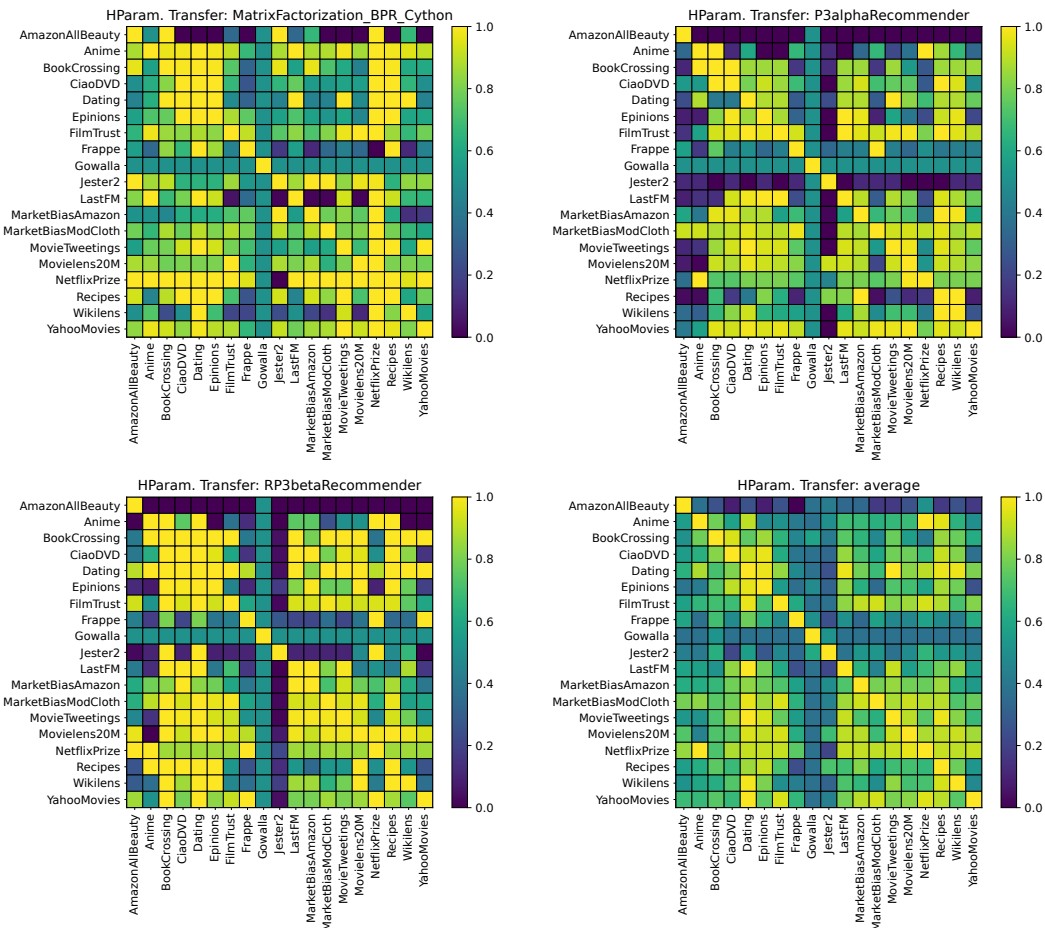

Figure 3: Transferability of hyperparameters across datasets, for three different algorithms, and the average of all algorithms (bottom right). For each plot, row $i$, column $j$ denotes the relative performance of an algorithm tuned on dataset $i$ and then evaluated on dataset $j$. A value close to 1 indicates that the hyperparameters transfer well from $i$ to $j$, while a value close to 0 indicates that the hyperparameters transfer poorly.

all algorithms, we see that it is particularly hard for hyperparameters to transfer to and from some datasets such as Gowalla and Jester2. Furthermore, the majority of pairs of datasets do not have strong hyperparameter transfer. Overall, these experiments give evidence that tuning the hyperparameters of an algorithm on one dataset and transferring to another dataset does not give high performance, motivating RecZilla which predicts the best hyperparameters for a given algorithm and dataset.

## 2.2 On the predictability of rec-sys algorithms

*Can attributes of the rec-sys dataset be used to predict the performance of rec-sys algorithms?*

**Dataset meta-features.** We calculate 383 different meta-features to characterize each dataset. These meta-features include statistics on the rating matrix—including basic statistics, the distribution-based features of Cunha et al. [23], and landmark features [24]—which measure the performance of simple rec-sys algorithms on a subset of the training dataset. Since these meta-features are used for algorithm selection, they are calculated using only the training split of each dataset. For more details on the dataset meta-features, see Appendix A.4.

**Algorithm performance prediction.** Table 2 shows the meta-features that are most highly-correlated with the performance (PREC@10) of each algorithm, using their default parameters.

Table 2: Highest absolute correlations computed across 85 datasets and weighed inversely proportional to dataset family frequency, over all pairs of algorithm families and meta-features, for the PREC@10 performance metric and the default algorithm hyperparameters.

| Abs. Correlation | Algorithm Family | Meta-feature |
|---|---|---|
| 0.941 | SlopeOne | Mean of item rating count distribution |
| 0.933 | CoClustering | Median of item rating count distribution |
| 0.887 | MF-BPR | Sparsity of rating matrix |
| 0.855 | RP3beta | Mean of item rating count distribution |
| 0.846 | UserKNN | Landmarker, Pure SVD, mAP@5 |

Several meta-features aare highly-correlated with algorithm performance; one of the simplest metrics—the mean of the item rating count distribution—is highly correlated with performance of two rec-sys algorithms. This experiment motivates the design of RecZilla in the next section, which trains a model using dataset meta-features to predict the performance of algorithms on new datasets.

As a toy-model version of RecZilla, we train three different meta-learner functions (XGBoost, KNN, and linear regression) using our meta-dataset, to predict performance metric PREC@10 for 10 rec-sys algorithms with high average performance (see Appendix A for details). We use leave-one-out evaluation for each meta-learner: one dataset family is held out for testing, while $m$ are used for training. Figure 4 shows the mean absolute error (MAE) of each meta-learner; these results are aggregated over 200 random samples of randomly-selected training dataset families. MAE decreases as more dataset families are added, suggesting that it is possible to estimate rec-sys algorithm performance using dataset meta-features.

We also find that performance metrics are not the only values that can be predicted with dataset meta-features. In particular, we find that the *runtime* of rec-sys algorithms is also highly correlated with different meta-features. Furthermore, we compute a simple measure of *dataset hardness*, which we compute as, given a performance metric, the maximum value achieved for that dataset across all algorithms. For example, if all 20 algorithms do not perform well on the MovieTweetings dataset, then we can expect that the MovieTweetings dataset is "hard". Once again, we find that certain dataset meta-features are highly correlated with dataset hardness. For more details on meta-feature correlation with algorithm runtimes and dataset hardness, see Appendix A.

The fact that dataset meta-features are correlated with algorithm performance, algorithm runtimes, and dataset hardness is a strong positive signal that meta-learning is worthwhile and useful in the context of recommender systems. We explore this direction further in the next section.

## 3   RecZilla: Automated Algorithm Selection

In the previous section, we found that *(1)* the best algorithm and hyperparameters strongly depend on the dataset and user-chosen performance metric, and *(2)* the performance of algorithms can be predicted from dataset meta-features. Points *(1)* and *(2)* naturally motivate an algorithm selection approach to rec-sys powered by meta-learning.

In this section, we present *RecZilla*, which is motivated by a practical challenge: given a performance metric and a new rec-sys dataset, quickly identify an algorithm and hyperparameters that perform well on this dataset. This challenge arises in many settings, e.g., when selecting good baseline algorithms for academic research, or when developing high-performing rec-sys algorithms for a commercial application. We begin with an overview and then formally present our approach.

**Overview.**   *RecZilla* is an algorithm selection approach powered by meta-learning. We use the results from the previous section as the meta-training dataset. Given a user-specified performance metric, we train a meta-model that predicts the performance of each of a set of algorithms and hyperparameters on a dataset, by using meta-features of the dataset. Given a new, unseen dataset, we compute the meta-features of the dataset, and then use the meta-model to predict the performance of each algorithm, returning the best algorithm according to the performance metric. See Figure 1.

**Preliminaries.** We start with notation for the rec-sys problem. Let $D$ denote a rec-sys dataset, consisting of a set of user-item interactions, split into a training and validation set. Let $a$ denote a rec-sys algorithm parameterized by a set of $k(a)$ hyperparameters $\boldsymbol{\omega} \in H(a) \subseteq \mathbb{R}^{k(a)}$, which is algorithm-dependent. Suppose we train algorithm $a$ on the training split of dataset $D$, using hyperparameters $\boldsymbol{\omega}$; we denote the *performance* of algorithm $a$ with hyperparameters $\boldsymbol{\omega}$ on dataset $D$ as $\phi(a, \boldsymbol{\omega}, D) \in \mathbb{R}$. The function $\phi(\cdot, \cdot, \cdot)$ represents a *performance metric* for the recommender system that is selected by the user; throughout this paper we refer to this function and its numerical value simply as *performance*. In this paper, larger values of $\phi(\cdot, \cdot, \cdot)$ indicates better performance. We report the *normalized* performance defined as

$$\overline{\phi}(a, \boldsymbol{\omega}, D) \equiv 100 \times \frac{\phi(a, \boldsymbol{\omega}, D) - P_D^{min}}{P_D^{max} - P_D^{min}},$$

where $P_D^{max}$ (resp. $P_D^{min}$) are the max (resp. min) performance of any algorithm on $D$.

Next we define notation for the meta-learning problem. Given a fixed performance function $\phi$, let $\mathcal{M} = \{(D_i, a_i, \boldsymbol{\omega}_i, y_i)\}_{i=1}^{M}$ denote a *meta-dataset* consisting of $M$ tuples. Each tuple con-

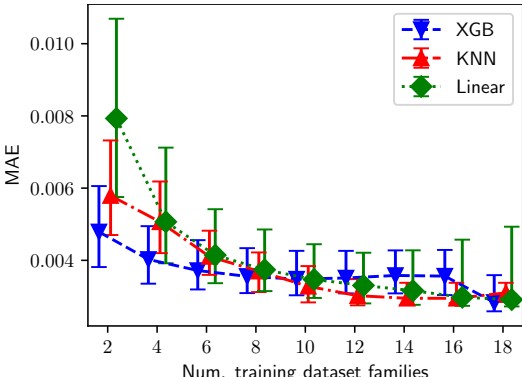

Figure 4: Three basic meta-learners (KNN, linear regression, and XGB) are trained randomly-selected dataset families to predict performance metric PREC@10. As more dataset families are added, the meta-learners are better able to predict rec-sys algorithm performance, suggesting that our dataset meta-features are useful for predicting rec-sys algorithm performance. Vertical axis shows mean absolute error (MAE), over all folds of leave-one-out validation, and 200 random trials; in each trial a different set of training datasets are chosen. Error bars show the 40th and 60th percentile.

sists of a dataset $D$, algorithm $a$ parameterized by $\boldsymbol{\omega}$, and performance $y$, where $y_i \equiv \phi(a_i, \boldsymbol{\omega}_i, D_i)$. Since many algorithms have a wide range of hyperparameters, we refer to the set of *parameterized algorithms* in the meta-dataset as $\mathcal{S} = \{(a_i, \boldsymbol{\omega}_i)\}_{i=1}^{N}$. We represent each dataset using vector $\boldsymbol{d} \in \mathbb{R}^m$, where each element of $\boldsymbol{d}$ is a meta-feature of the dataset. In this setting, the meta-learning task is to identify a function that estimates the performance of parameterized algorithm $(a, \boldsymbol{\omega})$ on dataset $D$.

### 3.1 The RecZilla Algorithm Selection Pipeline

The RecZilla algorithm selection pipeline takes as input a meta-dataset $\mathcal{M}$ built with a user-chosen performance function for the recommendation task $\phi(\cdot, \cdot, \cdot)$. RecZilla can accommodate any performance function that is a computable function of the performance metrics present in the meta-dataset. The RecZilla pipeline proposed here returns both an algorithm $a \in \mathcal{A}$ *and* a set of hyperparameters $\boldsymbol{\omega} \in H(a)$, so that RecZilla can be used without additional hyperparameter optimization.

Since the meta-learning problem is in the low-data regime, to guard against overfitting, we select a subset of the algorithms that have good coverage over the training dataset. We similarly select only the meta-features which are most predictive of the user-selected performance metric for the selected algorithms. Below we outline each of the steps used to build the proposed RecZilla pipeline.

1. **Algorithm subset selection:** We select a subset $\mathcal{S}' \subseteq \mathcal{S}$ of $n$ parameterized algorithms that collectively perform well on all datasets in the meta-training dataset $\mathcal{M}$, according to performance function $\phi$. We aim to select an algorithm subset with high *coverage* over the set of known datasets, where coverage of a subset $\mathcal{S}'$ is defined as

$$C(\mathcal{S}') = \frac{1}{|\mathcal{D}|} \sum_{D \in \mathcal{D}} \max_{(a, \boldsymbol{\omega}) \in \mathcal{S}'} \overline{\phi}(a, \boldsymbol{\omega}, D). \tag{1}$$

In other words, the coverage of subset $\mathcal{S}'$ is the normalized performance metric of the *best* performing algorithm in $\mathcal{S}'$, averaged over all datasets $\mathcal{D}$. Selecting a subset with maximum coverage is itself a difficult problem; we use a greedy heuristic as follows. We begin with $\mathcal{S}' = \{\}$ and iteratively add the parameterized algorithm $(a^*, \boldsymbol{\omega}^*) \in \arg\max_{(a, \boldsymbol{\omega}) \in \mathcal{S}} C(\mathcal{S}' \cup \{(a, \boldsymbol{\omega})\})$ to $\mathcal{S}'$ until $|\mathcal{S}'| = n$. That is, we greedily ensure that the coverage is maximized at each step.

Table 3: Comparison between RecZilla and two representative algorithm selection approaches from prior work. To give a fair comparison, the approaches are given the same meta-training datasets. We compute %Diff as defined in Section 3.2, as well as the Precision@10 for the predicted best algorithm. We report the mean and standard deviation across 50 trials for 19 test sets, for 950 total trials. The runtime is the average time it takes to output predictions on the meta-test dataset.

| Approach | Runtime (sec) | %Diff ($\downarrow$) | PREC@10 of best pred. ($\uparrow$) |
|---|---|---|---|
| `cunha2018` [27] | **0.39** | $52.9 \pm 23.0$ | $0.00813 \pm 0.0113$ |
| `cf4cf-meta` [25] | 6.68 | $43.5 \pm 21.8$ | $0.00808 \pm 0.00773$ |
| RecZilla | 6.69 | $\mathbf{33.2 \pm 22.8}$ | $\mathbf{0.00915 \pm 0.00840}$ |

2. **Meta-feature selection:** Similar to the previous point, we select a subset of meta-features with good coverage over the meta-training dataset $\mathcal{M}$, where here coverage is defined in terms of the correlation between algorithm performance and each meta-feature (see Appendix A.4 for details). Let $M : \mathcal{D} \to \mathbb{R}^m$ denote the resulting function that maps a dataset to a vector of meta-features.

3. **Meta-learning:** We learn a function $f : \mathbb{R}^m \to \mathbb{R}^n$, where $f(\boldsymbol{d}) = \hat{\boldsymbol{y}}$ is a vector of the estimated performance metric of all parameterized algorithms in $\mathcal{S}$ on dataset meta-features $\boldsymbol{d}$.

Note that the RecZilla pipeline has two hyperparameters: $n$, the number of parameterized algorithms in $\mathcal{S}'$; and $m$, the number of dataset meta-features used by the meta-learner. In our experiments, we run an ablation study with both $n$ and $m$, as well as different functions $f$ for the meta-learning model.

**Using RecZilla for Algorithm Selection.** After developing the RecZilla pipeline, we use the following steps to select an algorithm for a new dataset $D'$:

1. Calculate $m$ meta-features of the dataset $\boldsymbol{d}' \leftarrow M(D')$.
2. Estimate the performance of all parameterized algorithms: $\boldsymbol{y}' \leftarrow f(\boldsymbol{d}')$.
3. Return the parameterized algorithm in $\mathcal{S}'$ with the best estimated performance.

### 3.2 Experiments

In this section, we evaluate the end-to-end RecZilla pipeline. We start by describing the specific versions of RecZilla used in our experiments. We use four different meta-learning functions within RecZilla: XGBoost [12], linear regression, $k$-nearest neighbors, and uniform random. For KNN, we set $k = 5$ and use the $L_2$ distance from the selected meta-features.

**Experimental setup.** Focusing on performance metric PREC@10, we build a meta-dataset $\mathcal{M}$ using all rec-sys datasets, algorithms, and meta-features described in Section 2. We first use the algorithm selection and meta-feature selection procedures described above to select $n = m = 100$ parameterized algorithms and meta-features. For all experiments, we use the best 10 parameterized algorithms selected during this process. We vary both the number of training meta-datapoints and meta-features; the datapoints and features are randomly selected over 50 random trials. All RecZilla meta-learners are evaluated using leave-one-dataset-out evaluation: we iteratively select each dataset *family* as the meta-test dataset, and run the full RecZilla pipeline using the remaining datasets as the meta-training data. Splitting on dataset families rather than datasets ensures that there is no test data leakage. Then for each dataset $D$ in the test set, we compare the performance metric of the predicted best parameterized algorithm $(a', \boldsymbol{\omega}')$ to the performance metric of the ground-truth best algorithm $y^*$, using the percentage-difference-from-best: $\%\texttt{Diff} = 100 \times (y^* - \phi(D, a', \boldsymbol{\omega}'))/y^*$. $\%\texttt{Diff}$ is between 0 and 100, and smaller values indicate better performance.

**Results and discussion.** In Table 3, we compare RecZilla with two prior algorithm selection approaches: `cunha2018` [27] and `cf4c4-meta` [25], which are a comprehensive depiction of all prior work (see Appendix B.4 for justification, and for details of the experiment). Furthermore, note that `cunha2018` has no open-source code, and `cf4c4-meta` only has code in R. We find that RecZilla outperforms the other two approaches in both %Diff and in terms of the PREC@10 value of the rec-sys algorithm outputted by each meta-learning algorithm.

In Appendix C.3, Figure 5 (left) shows `%Diff` vs. the size of the meta-training set, and Figure 5 (right) shows the results of an ablation study on the number of selected meta-features $m$. See Appendix C.3 for more details and discussion.

**Pre-trained RecZilla models.** We release pre-trained RecZilla models for PREC@10, NDCG@10, and Item-hit Coverage@10, trained with XGBoost on all 18 datasets, with algorithms $n = 10$ and meta-features $m = 10$. We also include a RecZilla model that predicts the Pareto-front of PREC@10 and model training time, so that users can select their desired trade-off between performance and runtime. Finally, we include a pipeline so that users can choose a metric from the list of 315 (or any desired combination of the 315 metrics) and train the resulting RecZilla model.

## 4 Conclusions, Limitations, and Broader Impact

In this work, we conducted the first large-scale study of rec-sys approaches: we compared 24 algorithms and 100 sets of hyperparameters across 85 datasets and 315 metrics. We showed that for a given performance metric, the best algorithm and hyperparameters highly depend on the dataset. We also find that various meta-features of the datasets are predictive of algorithmic performance and runtimes. Motivated by these findings, we created RecZilla, the first metric-independent, hyperparameter-aware algorithm selection approach to recommender systems. Through empirical evaluation, we show that given a user-defined metric, RecZilla effectively predicts high-performing algorithms and hyperparameters for new, unseen datasets, substantially reducing the need for human involvement. We not only release our code and pretrained RecZilla models, but we also release the raw experimental results so that users can train new RecZilla models on their own test metrics of interest. This codebase, which includes a unified API for 85 datasets and 24 algorithms, may be of independent interest.

**Limitations.** While our work progresses prior work along several axes, there are still avenues for improvement. First, the meta-learning problem in RecZilla is low-data. Although we added nearly all common rec-sys research datasets into RecZilla, the result is still only 85 meta-datapoints (datasets). While we guarded against over-fitting to the training data in numerous ways, RecZilla can still be improved by more training data. Therefore, as new recommender system datasets are released in the future, our hope is to add them to our API, so that RecZilla continuously improves over time. Similarly, our hope is to add the most recent high-performing rec-sys approaches to our work, as well as algorithms released in the future. This includes adding neural network-based approaches, in addition to the six that we have already included. Another limitation is that RecZilla does not directly predict the performance of hyperparameters for algorithms on a given dataset. Although care must be taken to not overfit, modifying RecZilla to predict the performance of an algorithm together with a set of hyperparameters is an interesting avenue for future work. Finally, the magnitude of our evaluation ($84\,850$ models trained) leaves our meta-data susceptible to biases based on experiment success/failures. While we fixed many common errors such as out-of-memory errors, it was infeasible to give each experiment specific attention. Therefore, RecZilla may have higher uncertainty for the datasets and algorithms that are more likely to fail. An interesting future improvement to RecZilla would be to predict the likelihood that a new dataset will successfully train on a new dataset.

**Broader impact.** Our work is "meta-research": there is not one specific application that we target, but our work makes it substantially easier for researchers and practitioners to quickly train recommender system models when given a new dataset. On the research side, this is a net positive because researchers can much more easily include baselines, comparisons, and run experiments on large numbers of datasets, all of which lead to more principled empirical comparisons. On the applied side, our day-to-day lives are becoming more and more influenced by recommendations generated from machine learning models, which comes with pros and cons. These recommendations connect users with needed items that they would have had to spend time searching for [54]. Although these recommendations may lead to harmful effects such as echo chambers [38, 55], techniques to identify and mitigate harms are improving [42, 69].

## Acknowledgments and Disclosure of Funding

This work was supported by a GEM Fellowship, NSF CAREER Award IIS-1846237, NIST MSE Award #20126334, DARPA GARD #HR00112020007, and DoD WHS Award #HQ003420F0035.

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
