# A Experiment Details

This appendix outlines the algorithms, datasets, metrics, and hyperparameter selection used in RecZilla, as well as the details of our procedure for generating the meta-dataset. This codebase is publicly available[2], and is written in Python. The RecZilla codebase builds on another public Github repository[3].

## A.1 Generating Meta-Datasets

We generate the meta-dataset for reczilla using 24 rec-sys algorithms and 85 datasets. We use a leave-one-out training/validation split for each dataset: for each user, the last interaction is held out for validation, and all remaining interactions are used for training. For non-neural-network algorithms, each algorithm-dataset pair is given a 10 hour time limit for training and validation and trains/validates with up to 100 random hyperparameter sets (see Appendix A.5), on a single a `n1-highmem-2` instance on Google Cloud (2 vCPUs, 13GB memory). Neural-network based methods are trained using the default hyperparameters from their respective papers—meaning we do not sample different hyperparameter sets. For each dataset, neural-network methods are run for up to 15 hours on an `n1-highmem-16` Google Cloud instance (16 vCPUs, 104GB memory), with an NVIDIA Tesla T4 GPU. All neural methods use batch size 64, and are trained for up to 100 epochs; early stoppig occurrs if loss does not improve in 5 epochs. During validation, we calculate 21 different performance metrics at 15 different cutoffs, for a total of 315 different metrics (see Appendix A.6). Out of all 1 530 dataset-algorithm combinations tested in our experiments, 1 404 of them completed the train/validation procedure with at least one hyperparameter set within the 10 hour time limit. Most failed experiments failed due to invalid hyperparameter values, and some failed due to memory errors.

Algorithm runtime varied substantially across algorithm family and dataset. Table 4 shows runtime statistics over all experiments and all algorithms, for experiments that completed within the 10-hour time limit.

## A.2 Rec-sys Algorithms Implemented in RecZilla

Our experiments use 24 rec-sys algorithms. Algorithms with hyperparameters are associated with a hyperparameter space, as well as a set of "default" hyperparameters. Table 5 lists each implemented algorithm, along with its hyperparameter space and default parameters.

In the current implementation, we define several versions of User-KNN and Item-KNN, with one version for each similarity metric. This is done for convenience, since different KNN similarity metrics are associated with different hyperparameters. However, in our experiment results we treat all versions of User-KNN and Item-KNN as the same algorithm.

All algorithms here use the interface from [28, 29] (their codebase is publicly available[4]). All but two of our 24 algorithms use the implementation from this codebase; two algorithms (CoClustering and SlopeOne) use the implementation of Surprise [53], which is also publicly available.[5]

Table 5: Description of all algorithms implemented in RecZilla.

| Algorithm Name | Reference/ Description | Hyperparameter Space | |
|---|---|---|---|
| CoClustering | Clusters users and items. Uses their average ratings to predict new ratings. [39] [53] | num-control-users
num-control-items | : Int(1, 1000)
: Int(1, 1000) |

---

[2]`https://github.com/naszilla/reczilla`

[3]`https://github.com/MaurizioFD/RecSys2019_DeepLearning_Evaluation`

[4]`https://github.com/MaurizioFD/RecSys2019_DeepLearning_Evaluation/`

[5]`http://surpriselib.com/`

| | | | |
|---|---|---|---|
| EASE-R | Linear model designed for sparse data. Simplified version of an autoencoder [78]. | l2-norm | : [1, 1e7] |
| GlobalEffects | Rating predictions are based on a global score for each item and each user. | - | |
| iALS | Matrix factorization method. Leverages alternating least squares for optimization and uses regularization [51]. | num-factors
confidence-scaling
alpha
epsilon
reg | : Int(1, 200)
: {lin., log.}
: [1e-3, 50]
: [1e-3, 10]
: [1e-5, 1e-2] |
| ItemKNN-Asymmetric | k-nearest neighbors, item-based. [28, 76] Similarity between items is calculated using the asymmetric cosine similarity [5]. | top-K
shrink
alpha | : Int(5, 1000)
: Int(0, 1000)
: [0, 2] |
| ItemKNN-Cosine | k-nearest neighbors, item-based. [28, 76] Similarity between items is calculated using the cosine similarity. | top-K
shrink
normalize
feature-weighting | : Int(5, 1000)
: Int(0, 1000)
: Bool
: {none, BM25, TF-IDF} |
| ItemKNN-Dice | k-nearest neighbors, item-based. [28, 76] Similarity between items is calculated using the Sørensen-Dice coefficient. [32] | top-K
shrink
normalize | : Int(5, 1000)
: Int(0, 1000)
: Bool |
| ItemKNN-Euclidean | k-nearest neighbors, item-based. [28, 76] Similarity between items is calculated using the euclidean distance (l2 distance). | top-K
shrink
normalize
normalize-avg-row
similarity-from-distance | : Int(5, 1000)
: Int(0, 1000)
: Bool
: Bool
: {lin., log., exp.} |
| ItemKNN-Jaccard | k-nearest neighbors, item-based. [28, 76] Similarity between items is calculated using the Jaccard index. [70] | Same as ItemKNN-Dice | |
| ItemKNN-Tversky | k-nearest neighbors, item-based. [28, 76] Similarity between items is calculated using the Tversky index. [79] | top-K
shrink
alpha
beta | : Int(5, 1000)
: Int(0, 1000)
: [0, 2]
: [0, 2] |
| MF-AsySVD | Matrix factorization model that replaces user factors with the factors of items rated by that user. Items have multiple corresponding factors. [56] | sgd-mode
use-bias
num-factors
item-reg
user-reg
learning-rate
negative-interactions-quota | : {sgd, adagrad, adam}
: Bool
: Int(1, 200)
: [1e-5, 1e-2]
: [1e-5, 1e-2]
: [1e-4, 1e-1]
: [0, 0.5] |

| | | | |
|---|---|---|---|
| MF-BPR | Uses the Bayesian Personalized Ranking loss to learn a matrix factorization model [73]. | sgd-mode
num-factors
batch-size
positive-reg
negative-reg
learning-rate | : {sgd, adagrad, adam}
: Int(1, 200)
: {1, 2, 4, 8, 16, 32, 64, 128, 256, 512, 1024}
: [1e-5, 1e-2]
: [1e-5, 1e-2]
: [1e-4, 1e-1] |
| MF-FunkSVD | A modified version of the matrix factorization algorithm proposed in a blog post[6] [28]. | sgd-mode
use-bias
batch-size
num-factors
item-reg
user-reg
learning-rate
negative-interactions-quota | : {sgd, adagrad, adam}
: Bool
: {1, 2, 4, 8, 16, 32, 64, 128, 256, 512, 1024}
: Int(1, 200)
: [1e-5, 1e-2]
: [1e-5, 1e-2]
: [1e-4 1e-1]
: [0, 0.5] |
| NMF | Non-negative matrix factorization [22]. | num-factors
solver
init-type
beta-loss | : Int(1, 350)
: {coordinate-descent, multiplicative-update.}
: {random, nndsvda}
: {frobenius, kullback-leibler} |
| P3alpha | Computes the relevance between users and items based on random walks in a graph containing both users and items [20]. | top-K
alpha
normalize-similarity | : Int(5, 1000)
: [0, 2]
: Bool |
| PureSVD | Matrix factorization method based on SVD. | num-factors | : Int(1, 200) |
| Random | Predicts random ratings. | - | |
| RP3beta | Similar to P3alpha, but uses a reweighing scheme to compensate for item popularity [72]. | top-K
alpha
beta
normalize-similarity | : Int(5, 1000)
: [0, 2]
: [0, 2]
: Bool |
| SLIM-BPR | Uses a Sparse Linear Method (SLIM) optimized for Bayesian Personalized Ranking (BPR) loss. [7, 28] | top-K
symmetric
sgd-mode
lambda-i
lambda-j
learning-rate | : Int(5, 1000)
: Bool
: {sgd, adagrad, adam}
: [1e-5, 1e-2]
: [1e-5, 1e-2]
: [1e-4, 1e-1] |
| SLIMElasticNet | Sparse Linear Method (SLIM) [28, 62] | top-K
symmetric
l1-ratio
alpha | : Int(5, 1000)
: Bool
: [1e-5, 1]
: [1e-3, 1e-2] |
| SlopeOne | Uses linear functions to predict ratings for an item based on those from other items. [61] [53] | - | |
| TopPop | Recommends items based on global popularity regardless of user. | - | |

---

[6]https://sifter.org/~simon/journal/20061211.html

| | | |
|---|---|---|
| UserKNN-Asymmetric | k-nearest neighbors, item-based, using the asymmetric cosine similarity. [28, 74] | Same as ItemKNN-Asymmetric |
| UserKNN-Cosine | k-nearest neighbors, user-based, using the cosine similarity. [28, 74] | Same as ItemKNN-Cosine |
| UserKNN-Dice | k-nearest neighbors, user-based, using the Sørensen-Dice coefficient. [28, 74] | Same as ItemKNN-Dice |
| UserKNN-Euclidean | k-nearest neighbors, user-based, using the euclidean distance. [28, 74] | Same as ItemKNN-Euclidean |
| UserKNN-Jaccard | k-nearest neighbors, user-based, using the Jaccard index. [28, 74] | Same as ItemKNN-Jaccard |
| UserKNN-Tversky | k-nearest neighbors, user-based, using the Tversky index. [28, 74] | Same as ItemKNN-Tversky |

## A.3 RecZilla Datasets

The RecZilla codebase implements 88 datasets (3 additional datasets were added after our experiments on 85 datasets), derived from 20 dataset families. All datasets are listed in Table 6.

Table 6: Summary of datasets used to train and evaluate RecZilla.

| Dataset Name | # Interactions | # Items | # Users | Density |
|---|---|---|---|---|
| AmazonAllBeauty | 232 | 6,357 | 139 | 2.60E-04 |
| AmazonAllElectronics | 235 | 7,437 | 124 | 2.50E-04 |
| AmazonAlternativeRock | 328 | 3,842 | 120 | 7.10E-04 |
| AmazonAmazonFashion | 331 | 20,800 | 253 | 6.30E-05 |
| AmazonAmazonInstantVideo | 75,673 | 23,965 | 29,756 | 1.10E-04 |
| AmazonAppliances | 2,252 | 11,402 | 1,581 | 1.20E-04 |
| AmazonAppsforAndroid | 840,985 | 61,275 | 240,933 | 5.70E-05 |
| AmazonAppstoreforAndroid | 19 | 152 | 16 | 7.80E-03 |
| AmazonArtsCraftsSewing | 97,022 | 112,334 | 30,712 | 2.80E-05 |
| AmazonAutomotive | 300,532 | 320,112 | 100,163 | 9.40E-06 |
| AmazonBaby | 236,392 | 64,426 | 71,826 | 5.10E-05 |
| AmazonBabyProducts | 10,481 | 9,475 | 5,327 | 2.10E-04 |
| AmazonBeauty | 489,929 | 249,274 | 146,995 | 1.30E-05 |
| AmazonBlues | 98 | 896 | 25 | 4.40E-03 |
| AmazonBooks | 11,498,997 | 2,330,066 | 1,686,577 | 2.90E-06 |
| AmazonBuyaKindle | 6,312 | 1,858 | 2,715 | 1.30E-03 |
| AmazonCDsVinyl | 1,705,140 | 486,360 | 245,080 | 1.40E-05 |
| AmazonCellPhonesAccessories | 588508 | 319,678 | 245,110 | 7.51E-06 |
| AmazonChristian | 1,155 | 7,512 | 428 | 3.60E-04 |
| AmazonClassical | 528 | 2,301 | 152 | 1.50E-03 |
| AmazonClothingShoesJewelry | 1,615,940 | 1,136,004 | 496,837 | 2.86E-06 |
| AmazonCollectiblesFineArt | 1,066 | 5,705 | 230 | 8.12E-04 |
| AmazonComputers | 51 | 4,266 | 26 | 4.60E-04 |
| AmazonCountry | 151 | 1,677 | 47 | 1.90E-03 |
| AmazonDanceElectronic | 686 | 4,763 | 211 | 6.80E-04 |
| AmazonDavis | 38 | 58 | 28 | 2.30E-02 |
| AmazonDigitalMusic | 238,151 | 266,414 | 56,814 | 1.60E-05 |
| AmazonElectronics | 2,302,922 | 476,002 | 651,680 | 7.40E-06 |
| AmazonFolk | 236 | 2,366 | 38 | 2.60E-03 |
| AmazonGiftCards | 237 | 345 | 144 | 4.80E-03 |

| | | | | |
|---|---|---|---|---|
| AmazonGospel | 105 | 1,616 | 45 | 1.40E-03 |
| AmazonGroceryGourmetFood | 335,994 | 166,049 | 86,400 | 2.30E-05 |
| AmazonHardRockMetal | 156 | 1,063 | 41 | 3.60E-03 |
| AmazonHealthPersonalCare | 661,968 | 252,331 | 205,704 | 1.30E-05 |
| AmazonHomeImprovement | 45 | 3,855 | 32 | 3.60E-04 |
| AmazonHomeKitchen | 1,029,164 | 410,243 | 327,439 | 7.70E-06 |
| AmazonIndustrialScientific | 16,784 | 45,383 | 7,779 | 4.75E-05 |
| AmazonInternational | 608 | 5,544 | 193 | 5.70E-04 |
| AmazonJazz | 490 | 2,917 | 109 | 1.50E-03 |
| AmazonKindleStore | 1,387,653 | 430,530 | 213,192 | 1.50E-05 |
| AmazonKitchenDining | 81 | 3,658 | 63 | 3.50E-04 |
| AmazonLatinMusic | 21 | 613 | 13 | 2.60E-03 |
| AmazonLuxuryBeauty | 1,564 | 1,798 | 717 | 1.20E-03 |
| AmazonMagazineSubscriptions | 1,257 | 1,422 | 560 | 1.58E-03 |
| AmazonMiscellaneous | 416 | 5,262 | 164 | 4.80E-04 |
| AmazonMoviesTV | 1,894,519 | 200,941 | 319,406 | 3.00E-05 |
| AmazonMP3PlayersAccessories | 19 | 1,657 | 14 | 8.19E-04 |
| AmazonMusicalInstruments | 92,628 | 83,046 | 29,040 | 3.80E-05 |
| AmazonNewAge | 132 | 1,276 | 44 | 2.40E-03 |
| AmazonOfficeProducts | 166,878 | 130,006 | 59,858 | 2.10E-05 |
| AmazonOfficeSchoolSupplies | 41 | 3,229 | 21 | 6.05E-04 |
| AmazonPatioLawnGarden | 134,727 | 105,984 | 54,196 | 2.30E-05 |
| AmazonPetSupplies | 291,543 | 103,288 | 93,336 | 3.00E-05 |
| AmazonPop | 435 | 5,622 | 118 | 6.60E-04 |
| AmazonPurchaseCircles | 17 | 33 | 11 | 4.70E-02 |
| AmazonRapHipHop | 32 | 779 | 19 | 2.20E-03 |
| AmazonRB | 136 | 2,253 | 69 | 8.70E-04 |
| AmazonRock | 519 | 4,464 | 97 | 1.20E-03 |
| AmazonSoftware | 29,434 | 18,187 | 9,097 | 1.80E-04 |
| AmazonSportsOutdoors | 751,440 | 478,898 | 238,090 | 6.60E-06 |
| AmazonToolsHomeImprovement | 751,440 412,401 | 260,659 | 132,013 | 1.20E-05 |
| AmazonToysGames | 549,347 | 327,698 | 164,590 | 1.00E-05 |
| AmazonVideoGames | 308,086 | 50,210 | 84,273 | 7.30E-05 |
| AmazonWine | 215 | 1,228 | 84 | 2.10E-03 |
| Anime | 7,669,090 | 11,200 | 69,521 | 9.80E-03 |
| BookCrossing | 323,443 | 340,556 | 22,568 | 4.20E-05 |
| CiaoDVD | 47,102 | 16,121 | 4,743 | 6.20E-04 |
| Dating | 17,088,628 | 168,791 | 135,359 | 7.50E-04 |
| Epinions | 592,236 | 139,738 | 28,487 | 1.50E-04 |
| FilmTrust | 32,586 | 2,071 | 1,336 | 1.20E-02 |
| Frappe | 17,022 | 4,082 | 777 | 5.40E-03 |
| GoogleLocalReviews | 4,867,954 | 3,116,785 | 818,824 | 1.90E-06 |
| Gowalla | 3,735,522 | 1,247,095 | 91,846 | 3.30E-05 |
| Jester2 | 1,640,712 | 140 | 56,333 | 2.10E-01 |
| LastFM | 89,058 | 17,632 | 1,883 | 2.70E-03 |
| MarketBiasAmazon | 32,511 | 9,560 | 20,335 | 1.70E-04 |
| MarketBiasModCloth | 40,633 | 1,020 | 6,866 | 5.80E-03 |
| Movielens100K | 98,114 | 1,682 | 943 | 6.20E-02 |
| Movielens10M | 9,833,849 | 10,680 | 69,878 | 1.30E-02 |
| Movielens1M | 986,002 | 3,882 | 6,039 | 4.20E-02 |
| Movielens20M | 19,723,277 | 27,278 | 138,493 | 5.20E-03 |
| MovielensHetrec2011 | 851,372 | 10,109 | 2,113 | 4.00E-02 |
| MovieTweetings | 808,662 | 38,018 | 31,917 | 6.70E-04 |
| NetflixPrize | 99,521,398 | 17,770 | 476,694 | 1.20E-02 |
| Recipes | 819,642 | 231,637 | 35,464 | 1.00E-04 |
| Wikilens | 26,316 | 5,111 | 275 | 1.90E-02 |
| YahooMovies | 195,947 | 11,916 | 7,642 | 2.15E-03 |
| YahooMusic | 77,764,403 | 98,213 | 1,647,758 | 4.81E-04 |

Table 4: Min, mean, and max runtime for each algorithm, over all experiments, for both training and evaluation. The rightmost column shows the number of experiments collected for each algorithm. These runtime statistics only include experiments that completed within the 10 hour time limit, so they are skewed to be small, and should be interpreted as general trends. GlobalEffects, SlopeOne, Random, and TopPop do not have hyperparameters and therefore completed a maximum of 85 experiments. Furthermore, we tested multiple distance metrics for Item-KNN and User-KNN, resulting in more experiments.

| | Training time (seconds) | | | Evaluation time (seconds) | | | Num. experiments |
| | min | mean | max | min | mean | max | size |
| Alg. family | | | | | | | |
|---|---|---|---|---|---|---|---|
| CoClustering | 0.02 | 167.45 | 25766.23 | 0.05 | 51.09 | 11393.36 | 5106 |
| EASE-R | <0.01 | 13.85 | 454.14 | 0.05 | 14.99 | 341.97 | 4376 |
| GlobalEffects | <0.01 | 0.43 | 12.73 | 0.05 | 639.58 | 8191.36 | 85 |
| iALS | 0.69 | 369.30 | 24283.54 | 0.05 | 10.26 | 3558.97 | 3502 |
| Item-KNN | <0.01 | 100.28 | 13617.51 | 0.05 | 159.03 | 8192.90 | 12847 |
| MF-AsySVD | 0.03 | 207.76 | 28463.53 | 0.05 | 40.37 | 13293.74 | 4254 |
| MF-BPR | 0.02 | 82.19 | 9635.26 | 0.06 | 69.99 | 12117.41 | 5659 |
| MF-FunkSVD | 0.02 | 172.61 | 15650.11 | 0.05 | 50.49 | 12793.54 | 4938 |
| NMF | 0.01 | 221.50 | 20369.45 | 0.06 | 87.42 | 11471.30 | 2957 |
| P3alpha | <0.01 | 78.17 | 6264.47 | 0.05 | 62.40 | 6563.46 | 5816 |
| SVD | <0.01 | 3.58 | 353.12 | 0.05 | 99.06 | 10393.31 | 6132 |
| RP3beta | <0.01 | 80.45 | 7043.17 | 0.05 | 61.24 | 7067.75 | 5900 |
| Random | <0.01 | 0.09 | 2.35 | 0.06 | 937.67 | 16529.64 | 85 |
| SLIME-lasticNet | 0.06 | 142.95 | 25731.74 | 0.05 | 11.63 | 1816.69 | 4706 |
| SLIM-BPR | 0.03 | 82.57 | 31063.87 | 0.05 | 21.89 | 1962.25 | 5176 |
| SlopeOne | 0.05 | 17.73 | 470.27 | 0.07 | 45.77 | 803.25 | 48 |
| TopPop | <0.01 | 0.13 | 3.66 | 0.05 | 626.50 | 7501.84 | 85 |
| User-KNN | <0.01 | 73.28 | 30975.03 | 0.05 | 158.07 | 6327.46 | 13097 |

## A.4 Dataset Meta-Features & Meta-Feature Selection

We calculate a total of 383 meta-features for each rec-sys dataset, consisting of a few general meta-features, meta-features describing the distribution of ratings, and performance of landmarkers.

For each dataset, we extract the number of users, number of items, number of ratings, and the ratio of items to users. Furthermore, following the approach outlined in [23], we compute the sparsity of the matrix of interactions, and we systematically obtain a series of meta-features based on different distributions that can be obtained by aggregating the ratings in several ways.

**Distribution meta-features** The distribution meta-features are obtained in two steps, as described in [23]. First, we obtain a distribution. We do this in one of seven ways. We either take all of the ratings at once or we aggreggate ratings for either items or users in one of three different ways: sum, count, or mean. For each of these seven distributions, we then compute ten different descriptive statistics: mean, maximum, minimum, standard deviation, median, mode, Gini index, skewness, kurtosis, and entropy. This results in 70 distribution meta-features.

**Landmarkers** For landmarkers, we evaluate the performance of several baseline algorithms on a subset of the training set. We first select the subsample. Next, we partition this subsample into two sets: a "sub-training set" and a "sub-validation set". We train each landmarker on the sub-training set and compute performance metrics on the sub-validation set. We compute the 19 performance metrics described in Section A.6, plus three more algorithm-independent metrics:

- **Items in Evaluation Set:** measures the fraction of items with at least one rating in the evaluation set. This metric is algorithm-independent and only serves to describe the dataset and its split.

- **Users in Evaluation Set:** measures the fraction of users with at least one rating in the evaluation set. This metric is algorithm-independent and only serves to describe the dataset and its split.
- **Item Coverage:** fraction of items that are ranked within the top $K$ for at least one user.

All 22 metrics are evaluated at cutoffs 1 and 5, to create the meta-features.

The subsampling scheme is designed to satisfy several constraints. We limit the number of users to 100 and the number of items to 250. We also need to ensure there are at least 2 items rated per user so that the subsequent data split (holding out one item rating per user) does not result in cold users. Furthermore, we ensure the number of items selected is at least 6 so that we can evaluate the performance metrics with the cutoff of 5.

We start the subsampling process by filtering by the users that have at least 2 ratings. We next filter by the items all those users rated. If this results in less than 6 items, we add a random sample of the remaining (cold) items back to the set to have 6 items in total. Next, if the number of users is larger than 100, we take a random subsample of 100 of them and filter by those users. As before, we filter by the items those users rated, and ensure this results in at least 6 items by adding random items back if needed. If the number of items is still greater than 250, we must build an item subsample that results in at least two ratings per user. For each user, we randomly choose two of the items the user rated, and we take the union of these item choices. If this results in less than 250 items, we take a random sample of the remaining items to make the total number of items equal to 250.

Once this subsample is built, we split the subsample into the sub-training and sub-validation set by leaving 1 random item out for each user. We are then ready to run our landmarkers on those sets.

Our landmarkers consist of TopPop, ItemKNN, UserKNN, and PureSVD. For ItemKNN and UserKNN, we use cosine similarity and set the number of neighbors $k$ to 1 and 5. For PureSVD, we use 1 and 5 for the number of latent factors. This results in a total of 7 landmarkers.

Running all 7 landmarkers and computing all 22 metrics at the 2 different cutoffs results in a total of 308 landmarker meta-features.

### A.5 Hyperparameter Sampling

For each algorithm with hyperparameters, we test up to 100 parameter sets, limited by the 10-hour time limit used in our experiments. The first evaluated hyperparameter set is the default hyperparameters [7] The remaining 99 hyperparameters are sampled using the ranges specified in Table 5, using Sobol sampling.

### A.6 Evaluation Metrics

Each of the evaluation metrics that we use during validation measures the quality of ranking based on the item ratings. For each user, we generate predicted ratings for all items and rank the items according to the predicted rating (in descending order). We trim these ranked lists at a given cutoff, which we denote by $K$. We then compute different metrics using these user-wise top $K$ items. Some metrics also consider the set of relevant items within these top $K$, defined as those for which the user rated the item in the evaluation set.

We use 21 different metrics, and we compute them using cutoffs in $\{1, 2, 3, 4, 5, 6, 7, 8, 9, 10, 15, 20, 30, 40, 50\}$. This results in 315 different metric/cutoff combinations total. Only a small subset of these metrics are used in our analysis; however, any user-chosen metric (or combination of metrics) can be used to define a performance function for RecZilla.

The metrics are calculated using the implementation in the public repository[8] that our codebase is built on. Below is a list of metrics calculated during model evaluation:

- **Average Popularity:** measures the popularity of the recommended items. The popularity for each item is the frequency with which it was rated in the training set. These popularities

---

[7]See https://github.com/naszilla/reczilla.
[8]https://github.com/MaurizioFD/RecSys2019_DeepLearning_Evaluation

are normalized by the largest popularity. Next, for each user, we compute the normalized popularity of each of the items within its top $K$ and take their mean. Finally, we average across all users.

- **Average Reciprocal Hit-Rank (ARHR):** similar to MRR, except the reciprocal ranks for all relevant items (not just the first) are summed together.

- **Diversity (Gini):** computes the Gini diversity index [9] of the global distribution of items ranked within the top $K$ across all users. Higher values indicate higher diversity.

- **Diversity (Herfindahl):** computes the Herfindahl index [2] of the global distribution of items ranked within the top $K$ across all users. Higher values indicate higher diversity.

- **Diversity (Shannon):** computes the Shannon entropy of the global distribution of items ranked within the top $K$ across all users.

- **F1 Score:** the harmonic mean between precision and recall.

- **Hit Rate:** fraction of users for which at least one relevant item is present within the top $K$.

- **Item Coverage (Hit):** fraction of relevant items that are ranked within the top $K$ for at least one user.

- **Mean Average Precision (mAP):** the mean of the average precision across all users. The average precision for a user is computed as follows: for any position $i \leq K$ occupied by a relevant item, we compute the precision at $i$. We sum all of these precision values and divide the total by $K$.

- **Mean Average Precision - Min Den:** similar to mAP, but using a modified version of the average precision. If the number of test items for the user is smaller than $K$, we divide the sum (in the last step of the average precision computation) by this number instead of by $K$.

- **Mean Inter-List Diversity:** measures how different the top $K$ lists are for all users, as originally proposed in [86]. For each pair of users, we compute the fraction of items in their top $K$ items that are not present in both lists. Taking the average across all users yields the mean inter-list diversity. The codebase implements a more efficient but equivalent way of computing this metric.

- **Mean Reciprocal Rank (MRR):** the mean of the reciprocal rank across all users. The reciprocal rank for a user is the reciprocal of the rank of the most highly-ranked relevant item or 0 if there is none.

- **Normalized Discounted Cumulative Gain (NDCG):** first, the discounted cumulative gain (DCG) of the top $K$ ranking is computed by adding, for all relevant items in the top $K$, a gain discounted logarithmically in terms of the rank. The NDCG is obtained by dividing the DCG of the ranking by that of an ideal ranking (a ranking ordered by relevance).

- **Novelty:** a metric that rewards recommending items that were not popular in the training set [86]. For each item, we compute the fraction of ratings in the training set that correspond to the item. The novelty contributed by the item is computed by taking the negative logarithm of that fraction and dividing by the total number of items, so that items that were seldom seen in the training set result in high contributions. Now, for any user, we compute the novelty as the sum of these contributions for the top $K$ items. Finally, we average the metric across all users.

- **Precision:** the fraction of items in the top $K$ that are relevant, computed across all users.

- **Precision Recall Min:** similar to precision, but if there are less test items than $K$, the fraction is computed with respect to the number of test items.

- **Recall:** the fraction of relevant items that were placed within the top $K$, computed across all users.

- **User Coverage:** fraction of users both present in the evaluation set and for which the model is able to generate recommendations. In practice, all the algorithms used were able to generate recommendations for all users, since our splitting procedures did not result in cold users, so this metric was always 1 for our datasets and algorithms.

---

[9] https://www.statsdirect.com/help/default.htm#nonparametric_methods/gini.htm

Table 7: Highest absolute correlations between algorithm running time (default hyperparameters) and meta-features.

| Abs. Correlation | Algorithm Family | Meta-feature |
|---|---|---|
| 0.999 | MF-FunkSVD | Number of interactions |
| 0.997 | MF-BPR | Number of users |
| 0.993 | GlobalEffects | Number of interactions |
| 0.990 | TopPop | Number of interactions |
| 0.986 | ItemKNN | Kurtosis of item rating sum distribution |

Table 8: Highest absolute correlations between dataset hardness (negative of maximum PREC@10 achieved over all algorithms) and meta-features.

| Abs. Correlation | Pos. or Neg. Corr. | Meta-feature |
|---|---|---|
| 0.752 | Negative | Entropy of ratings |
| 0.668 | Negative | Mode of user rating count distribution |
| 0.655 | Negative | Landmarker, UserKNN ($k = 5$), Item Coverage @ 1 |
| 0.652 | Negative | Landmarker, TopPop, Recall @ 5 |
| 0.652 | Negative | Landmarker, TopPop, Hit Rate @ 5 |

- **User Coverage (Hit):** fraction of users both present in the evaluation set and for which the model is able to generate at least one relevant recommendation within the top $K$ items. It is equal to the product of the hit rate and the user coverage. Because the latter was always equal to 1, the user coverage (hit) was the same as the hit rate in our experiments.

### A.7 Additional details and experiments from Section 2.2

Recall that in Table 2, we showed the meta-features that are most highly correlated with the performance (PREC@10) of each algorithm, using their default parameters. In Table 7, we run the same analysis using "training time" instead of PREC@10 as the metric. We see that for some algorithms, the runtime is very highly correlated with certain meta-features such as "number of interactions".

Next, we compute a simple measure of *dataset hardness*, which we compute as, given a performance metric, the negative of the maximum value achieved for that dataset across all algorithms. For example, if all 20 algorithms do not perform well on the MovieTweetings dataset, then we can expect that the MovieTweetings dataset is "hard". In Table 8, we show the meta-features that are most highly correlated with the *hardness* of each algorithm, where hardness is calculated as -PREC@10. We find that the entropy of the rating matrix is most correlated with dataset hardness.

**Additional details from Section 2.2.** In this section, we give more details of the experimental setup of the experiment in Figure 4 from Section 2.2. We compare three different meta-learner functions: XGBoost, linear regression, and KNN with $k$=5 and $L_2$ distance. In order to give a fair comparison, we use a fixed set of 10 high-performing rec-sys algorithms, and also a fixed set of 10 meta-features that have high correlation with the performance metric PREC@10. The meta-learner models are trained to predict PREC@10 of all 10 algorithms.

We use leave-one-out evaluation for each meta-learner: one dataset family is held out for testing, while $x$ other dataset families are used for training, for $x$ from 2 to 20. For each meta-learner, we compute the mean absolute error (MAE): the absolute differences between the prediction and ground truth for each algorithm are averaged. These are then averaged over 200 trials of randomly-selected training dataset families.

## B  RecZilla Meta-Learning Pipeline

In this section, we give more details of the RecZilla pipeline, and we give an additional experiment in which we compare RecZilla to other existing rec-sys meta-learning approaches.

The RecZilla pipeline consists of the following components: initial algorithm selection, meta-feature selection, and finally the meta-learner for algorithm selection. The purpose of the first two components is to reduce the dimensionality of the dataset and reduce the risk of overfitting for the classifier. We describe each of these components in the following sections. In all that follows, we assume that the user provides (a) a performance metric function $\phi$, (b) the number of parameterized algorithms to be considered by the meta-learner $n$, and (c) the number of dataset meta-features to be considered by the meta-learner $m$. In addition, we assume access to a meta-dataset $\mathcal{M}$, such as the one described (and already pre-computed) in this paper.

## B.1    Initial algorithm selection

We first select $n$ parameterized algorithms which have high *coverage* over all datasets in meta-dataset $\mathcal{M}$ (see Section 3.1 of the paper). Since data is relatively scarce in this meta-learning task, we select a subset of $n$ algorithms to reduce the dimensionality of the meta-learner prediction target.

## B.2    Meta-feature selection

Since our meta-dataset $\mathcal{M}$ includes hundreds of features, we restrict our meta-learner to $m$ features to avoid over-fitting. This is the same approach taken by prior work [27]. It is computationally infeasible to find the set of $m = 10$ best meta-features out of a set of 383, since we would need to check $\binom{383}{10} \approx 10^{19}$ combinations of meta-features. Instead, we iteratively grow a set of $m$ meta-features which are highly correlated with the user-specified performance metric, without selecting redundant features, by using a "greedy" approach (similar in spirit to prior work [27]). Here we require that the (a) performance metric function $\phi$ is chosen ahead of time, and (b) a set of $n$ parameterized algorithms have been selected. We introduce some additional notation for this section to describe our meta-feature selection process. Let $\boldsymbol{y}^i \in \mathbb{R}^{|\mathcal{D}|}$ be the vector of performance metric $\phi$ for parameterized algorithm $i \in 1, \ldots, n$, for all datasets in $\mathcal{D}$. Let $\boldsymbol{d}^j \in \mathbb{R}^{|\mathcal{D}|}$ be the vector of meta-feature $j$ for all datasets in $\mathcal{D}$, and let $J$ be the total number of meta-features. Let $F$ denote a set of feature indices that corresponds to selected features.

For each (algorithm, meta-feature pair), we first compute the absolute value of the Pearson correlation between the meta-feature and the performance of each parameterized algorithm $i$, across all datasets: $c_{ij} \leftarrow |\text{corr}(\boldsymbol{y}^i, \boldsymbol{d}^j)|$ for all $i \in \{1, \ldots, n\} = [n]$ and $j \in \{1, \ldots, J\} = [J]$. When computing this correlation, each sample is weighed inverse-proportionally to the size of the dataset family it corresponds to—to prevent large dataset families (such as Amazon) from dominating the correlation computation.

We select the first feature by finding the largest absolute correlation coefficient between any of the meta-features and parameterized algorithms, and we choose the meta-feature corresponding to it:

$$j_0 \leftarrow \underset{j \in [J]}{\arg\max} \left( \max_{i \in [n]} c_{ij} \right).$$

All remaining $(m - 1)$ meta-features are selected such that we maximize the *improvement* in the absolute correlation between the selected features and the selected algorithms' performance. This way, we avoid selecting highly correlated features. Algorithm 1 gives a pseudocode description of this feature-selection process.

## B.3    Metalearner for algorithm selection

The goal of the metalearner is to predict the performance of all $n$ selected parameterized algorithms on a new dataset. The input to this meta-learner is the set of $m$ meta-features selected in the previous selection, and the output is an $n$-dimensional vector of performance metrics for all selected algorithms. We treat this as a multi output regression problem, and our experiments test three different models: a RegressiorChain with XGBoost as the base model, KNN with $k = 5$ and $L_2$ distance, and multinomial linear regression. For training these meta-learner models, we used the squared error cost function.

To train the meta-learner we construct a final meta-dataset consisting of one tuple $(\boldsymbol{d}, \boldsymbol{y})$ for each dataset represented in $\mathcal{M}^{train}$, where $\boldsymbol{d}$ is a vector of $m$ meta-features for the dataset, and $\boldsymbol{y}$ is a vector of the performance of $n$ parameterized rec-sys algorithms in the set of selected parameterized algorithms $\mathcal{S}'$.

**Algorithm 1** RecZilla Meta-feature selection

---

**Require:** $\boldsymbol{d}^j \in \mathbb{R}^{|\mathcal{D}|}, \forall j \in [J]$               $\triangleright$ $J$ vectors of meta-features for each dataset
**Require:** $\boldsymbol{y}^i \in \mathbb{R}^{|\mathcal{D}|}, \forall i \in [n]$        $\triangleright$ $n$ vectors of performance metrics for each algorithm
**Require:** $m > 0$                            $\triangleright$ number of features to select
    $F \leftarrow \{\}$                            $\triangleright$ indices of selected features
    $x_i = 0, \forall i \in [n]$         $\triangleright$ max abs. correlation between any selected meta-feature and $\boldsymbol{y}^i$
    $c_{ij} \leftarrow |\text{corr}(\boldsymbol{y}^i, \boldsymbol{d}^j)|, \forall i \in [n], j \in [J]$
    **while** $|F| < m$ **do**

$$j' \leftarrow \underset{j \in [J]}{\arg\max} \left[ \max_{i \in [n]} (c_{ij} - x_i) \right]$$

       $F \leftarrow F \cup \{j'\}$
       $x_i \leftarrow \max\{x_i, c_{ij'}\}, \forall i \in [n]$          $\triangleright$ update the max. abs. correlation for each alg.
    **end while**
    **return** $F$

---

### B.4    Additional details on comparisons to other algorithm selection methods for rec-sys

As described in section of Section 1, there are a few existing works on algorithm selection for recommender systems. In this section, we describe the two most relevant and recent works in more detail, and we give more details on the experiment in Section 3 that empirically compared RecZilla with existing works (Table 3).

Multiple approaches for algorithm selection for recommender systems were developed between 2011 and 2018 [3, 23, 35, 43, 52], but in 2018, Cunha et al. [27] gave a thorough empirical analysis of all recommender systems studied up to that point. They showed that the best approach was to use meta-feature selection over 74 distribution-based meta-features, and to use polynomial SVM as the meta-model. Their study used 38 total datasets. We refer to this algorithm as `cunha2018`.

Two more algorithm selection for rec-sys approaches were released after Cunha et al. [27]. CF4CF [26], which used subsampling landmarkers, and CF4CF-META, which combined CF4CF with the earlier meta-learning approaches described in the previous paragraph. Specifically, CF4CF-META uses landmarkers as well as the 74 distribution-based meta-features. It also uses KNN as the meta-model. Their study uses 38 total datasets. We refer to this approach as `cf4cf-meta`.

Given that `cunha2018` subsumes all work prior to 2018, and `cf4c4-meta` subsumes all subsequent work, we compare RecZilla against these two algorithms as a comprehensive depiction of all prior work. Note that `cunha2018` has no open-source code, and `cf4c4-meta` only has code in R. Furthermore, in order to give a more fair empirical study, we implement both approaches directly within our codebase. Additionally, to give a fair comparison, each model uses the same meta-training datasets, algorithm selection procedure, and base algorithms. Since a main novelty of RecZilla is predicting hyperparameters as well as algorithms, the other two approaches are only given the algorithms with the default hyperparameters.

We run an experiment with `cunha2018`, `cf4c4-meta`, and RecZilla, in the same setting as Section 3.2 (i.e., the setting of Figure 5), where we run leave-one-dataset-out evaluation and average the results over all test datasets. The algorithms are given all 18 dataset families not in the test set, to use as training data. We run 50 trials for all 19 possible test sets in the leave-one-dataset-out evaluation, for a total of 950 trials. See Table 3. RecZilla outperforms the other two approaches in both %Diff and in terms of the PREC@10 value of the rec-sys algorithm outputted by each meta-learning algorithm.

## C    Additional Results and Discussion

### C.1    Detailed algorithmic results

Tables 9, 10, and 11 show the min (best), max (worst) and mean rank for each algorithm, over all 85 datasets, over a subset of metrics. Most algorithms are ranked in the top-3 for each metric, on at least one dataset; furthermore, most algorithms are ranked *poorly* on at least one dataset. However,

average performance varies widely across algorithms; for example, Item-KNN is has an average rank at most 3 for most metrics, while TopPop, GLobalEffects, SlopeOne, have average ranks closer to 20.

## C.2 A guide to practitioners

In this section, we lay out the key takeaways and insights for practical recommender system design, so that practitioners can get the largest value from our work and use our work most effectively.

**Insights from Section 2.** In Section 2, we presented a large-scale empirical study of rec-sys algorithms, studying the generalizability and predictability of rec-sys algorithms. The key takeaway from this section for practitioners is that rec-sys algorithms are *not* generalizable (see Table 1 and Figure 2). Therefore, practitioners who have used an algorithm successfully on one rec-sys dataset will likely need to try other diverse types of algorithms when starting on a brand new dataset, even if using a neural network-based algorithm.

Practitioners may also use our analysis for intuition when working with RecZilla models, in a number of different ways. The particular use cases and goals of a practitioner may be very specific, but we give a few examples on how to leverage insights from Section 2 below.

- Table 8 gives properties of a dataset that are predictive of "dataset hardness." Of course, this is based on correlation alone, but it suggests that a dataset is "harder" than average if the `entropy of ratings` is greater than 2.07 (this value can be computed directly inside our codebase [10]). If less than 2.07, the practitioner can continue as usual, but if greater than 2.07, it may be desirable to check if the dataset is noisy and can be cleaned, or if more data can be obtained. Or, it could be a sign that the practitioner should expect to spend more time than usual tuning the recommender system model (after running RecZilla, of course).

- If the practitioner is concerned with the training time and latency of their recommender system model, then they can estimate the runtimes in advance. For example, according to Table 4, `number of interactions` has a very high correlation with the training time of `MF-FunkSVD`. By running a simple regression, we have the following formula:

$$\text{runtime of MF-FunkSVD} = 0.00439 \cdot (\text{number of interactions}) + 0.304.$$

  Similar formulas can be computed for other algorithms, by using our codebase [11].

- If the practitioner is interested in using a specific algorithm on a variety of datasets, then they can gain insights by looking up dataset meta-features that specifically have high correlation with the performance metric and runtime of that algorithm. For example, a practitioner might prefer to use the CoClustering algorithm for the interpretable clusters that it gives. By looking at Table 2, and by computing additional correlations using our codebase [11], the practitioner can find the insight that `Median of item rating count distribution` and `Median of item rating sum distribution` are both predictive of the performance of CoClustering, which may be useful to know as a sanity check when tuning and deploying CoClustering on different datasets.

**A guide for using RecZilla.** The goal of RecZilla is to allow practitioners to very quickly train a model that performs well, when faced with a new dataset. The practitioner may then choose to further tune the model to reach even stronger performance.

The first step for the practitioner is to define their objective. If the objective is PREC@10, NDCG@10, or Item-hit coverage@10, then the practitioner can use one of the pre-trained models included in our repository. First, follow the installation instructions in the README. [11] Next, follow the instructions in the `Main script overview` section of the README. Specifically, choose the `metamodel_filepath` that matches the desired objective.

If the desired objective is not one of the above three metrics, then the practitioner shoudl train a new meta-model. This can be done by following the instructions in the `Training a new meta-model`

---

[10]See https://github.com/naszilla/reczilla/blob/main/RecSys2019_DeepLearning_ Evaluation/Metafeatures/README.md and https://github.com/naszilla/reczilla/blob/ main/RecSys2019_DeepLearning_Evaluation/Metafeatures/DistributionFeatures.py

[11]https://github.com/naszilla/reczilla

Table 9: The performance of each rec-sys algorithm varies depending on the dataset and evaluation metric. Each row corresponds to a different metric (NDCG, precision, recall, and hit-rate) at a particular cutoff. Each algorithm (columns) are ranked according to each metric, values show the "min/max (mean)" rank over all datasets for which at least 10 algorithms produced a result. This table includes all 24 implemented algorithms. In this table, maximum rank is 24, and lower rank indicates better performance. This table is continued on the next page. (Part 1 of 3)

| | Item-KNN | User-KNN | RP3beta | RP3beta | iALS | EASE-R | SlopeOne | CoClustering |
|---|---|---|---|---|---|---|---|---|
| NDCG@1 | 1/4 (1.5) | 2/18 (11.1) | 2/12 (5.3) | 1/12 (3.7) | 1/19 (7.4) | 1/18 (4.2) | 7/22 (15.7) | 1/20 (14.1) |
| NDCG@2 | 1/10 (1.8) | 2/17 (9.3) | 2/14 (6.0) | 1/14 (4.0) | 1/19 (7.5) | 1/18 (4.8) | 9/23 (16.7) | 1/21 (14.9) |
| NDCG@5 | 1/12 (2.2) | 1/16 (7.4) | 2/14 (5.9) | 1/15 (4.0) | 2/19 (7.4) | 1/18 (5.2) | 12/23 (17.8) | 2/21 (15.5) |
| NDCG@10 | 1/13 (2.3) | 1/15 (6.6) | 2/15 (6.2) | 1/14 (4.3) | 1/19 (7.2) | 1/18 (5.2) | 14/23 (18.2) | 2/21 (15.7) |
| NDCG@50 | 1/13 (2.9) | 1/15 (5.9) | 2/15 (6.4) | 1/16 (4.9) | 1/19 (6.5) | 1/18 (5.8) | 13/23 (18.4) | 3/21 (15.8) |
| PRECISION@1 | 1/4 (1.5) | 2/18 (11.1) | 2/12 (5.3) | 1/12 (3.7) | 1/19 (7.4) | 1/18 (4.2) | 7/22 (15.7) | 1/20 (14.1) |
| PRECISION@2 | 1/11 (1.9) | 2/16 (8.4) | 1/14 (5.9) | 1/14 (3.9) | 1/19 (7.4) | 1/18 (4.6) | 7/23 (16.6) | 1/21 (14.7) |
| PRECISION@5 | 1/12 (2.6) | 1/16 (6.1) | 1/14 (5.9) | 1/13 (4.2) | 1/19 (7.1) | 1/18 (5.6) | 13/23 (17.9) | 3/21 (15.4) |
| PRECISION@10 | 1/12 (2.9) | 1/14 (4.9) | 1/14 (6.6) | 1/15 (4.6) | 1/19 (6.2) | 1/18 (6.6) | 14/23 (18.3) | 7/21 (15.9) |
| PRECISION@50 | 1/14 (4.3) | 1/12 (4.1) | 1/17 (7.3) | 1/18 (5.8) | 1/19 (5.4) | 2/18 (8.1) | 13/23 (18.1) | 2/21 (15.7) |
| RECALL@1 | 1/4 (1.5) | 2/18 (11.1) | 2/12 (5.3) | 1/12 (3.7) | 1/19 (7.4) | 1/18 (4.2) | 7/22 (15.7) | 1/20 (14.1) |
| RECALL@2 | 1/11 (1.9) | 2/16 (8.4) | 1/14 (5.9) | 1/14 (3.9) | 1/19 (7.4) | 1/18 (4.6) | 7/23 (16.6) | 1/21 (14.7) |
| RECALL@5 | 1/12 (2.6) | 1/16 (6.1) | 1/14 (5.9) | 1/13 (4.2) | 1/19 (7.1) | 1/18 (5.6) | 13/23 (17.9) | 3/21 (15.4) |
| RECALL@10 | 1/12 (2.9) | 1/14 (4.9) | 1/14 (6.6) | 1/15 (4.6) | 1/19 (6.2) | 1/18 (6.6) | 14/23 (18.3) | 7/21 (15.9) |
| RECALL@50 | 1/14 (4.3) | 1/12 (4.1) | 1/17 (7.3) | 1/18 (5.8) | 1/19 (5.4) | 2/18 (8.1) | 13/23 (18.1) | 2/21 (15.7) |
| HIT-RATE@1 | 1/4 (1.5) | 2/18 (11.1) | 2/12 (5.3) | 1/12 (3.7) | 1/19 (7.4) | 1/18 (4.2) | 7/22 (15.7) | 1/20 (14.1) |
| HIT-RATE@2 | 1/11 (1.9) | 2/16 (8.4) | 1/14 (5.9) | 1/14 (3.9) | 1/19 (7.4) | 1/18 (4.6) | 7/23 (16.6) | 1/21 (14.7) |
| HIT-RATE@5 | 1/12 (2.6) | 1/16 (6.1) | 1/14 (5.9) | 1/13 (4.2) | 1/19 (7.1) | 1/18 (5.6) | 13/23 (17.9) | 3/21 (15.4) |
| HIT-RATE@10 | 1/12 (2.9) | 1/14 (4.9) | 1/14 (6.6) | 1/15 (4.6) | 1/19 (6.2) | 1/18 (6.6) | 14/23 (18.3) | 7/21 (15.9) |
| HIT-RATE@50 | 1/14 (4.3) | 1/12 (4.1) | 1/17 (7.3) | 1/18 (5.8) | 1/19 (5.4) | 2/18 (8.1) | 13/23 (18.1) | 2/21 (15.7) |

Table 10: Continuation of Table 9 (Part 2 of 3).

| | MF-AsySVD | MF-FunkSVD | SVD | NMF | MF-BPR | SLIM-BPR | SLIM-ElasticNet |
|---|---|---|---|---|---|---|---|
| NDCG@1 | 2/17 (10.9) | 1/13 (8.5) | 1/13 (6.1) | 1/12 (7.0) | 1/17 (10.4) | 1/12 (4.1) | 1/17 (5.8) |
| NDCG@2 | 2/18 (11.3) | 1/17 (9.7) | 1/15 (6.0) | 2/13 (7.0) | 2/17 (11.0) | 1/13 (4.2) | 1/17 (6.2) |
| NDCG@5 | 4/18 (11.3) | 1/15 (10.0) | 2/12 (6.3) | 1/13 (7.6) | 3/17 (12.0) | 1/14 (4.2) | 1/17 (7.1) |
| NDCG@10 | 4/18 (11.3) | 2/15 (10.1) | 1/12 (6.4) | 1/13 (7.2) | 2/19 (12.0) | 1/14 (5.0) | 1/17 (7.1) |
| NDCG@50 | 5/18 (11.1) | 1/15 (10.1) | 1/16 (6.4) | 1/13 (7.0) | 2/17 (12.3) | 1/14 (6.1) | 1/16 (7.8) |
| PRECISION@1 | 2/17 (10.9) | 1/13 (8.5) | 1/13 (6.1) | 1/12 (7.0) | 1/17 (10.4) | 1/12 (4.1) | 1/17 (5.8) |
| PRECISION@2 | 1/18 (10.8) | 2/15 (9.5) | 1/14 (5.9) | 1/12 (6.7) | 2/16 (11.0) | 1/13 (3.8) | 1/17 (6.0) |
| PRECISION@5 | 1/18 (10.3) | 1/15 (9.3) | 1/11 (5.8) | 1/14 (7.2) | 1/17 (11.3) | 1/14 (4.7) | 1/17 (7.6) |
| PRECISION@10 | 1/18 (10.0) | 1/14 (9.4) | 1/11 (6.0) | 1/15 (6.9) | 1/18 (11.4) | 1/14 (5.3) | 1/17 (8.1) |
| PRECISION@50 | 1/16 (9.7) | 1/15 (9.0) | 1/16 (5.4) | 1/12 (6.8) | 1/17 (11.6) | 1/14 (7.3) | 1/16 (9.2) |
| RECALL@1 | 2/17 (10.9) | 1/13 (8.5) | 1/13 (6.1) | 1/12 (7.0) | 1/17 (10.4) | 1/12 (4.1) | 1/17 (5.8) |
| RECALL@2 | 1/18 (10.8) | 2/15 (9.5) | 1/14 (5.9) | 1/12 (6.7) | 2/16 (11.0) | 1/13 (3.8) | 1/17 (6.0) |
| RECALL@5 | 1/18 (10.3) | 1/15 (9.3) | 1/11 (5.8) | 1/14 (7.2) | 1/17 (11.3) | 1/14 (4.7) | 1/17 (7.6) |
| RECALL@10 | 1/18 (10.0) | 1/14 (9.4) | 1/11 (6.0) | 1/15 (6.9) | 1/18 (11.4) | 1/14 (5.3) | 1/17 (8.1) |
| RECALL@50 | 1/16 (9.7) | 1/15 (9.0) | 1/16 (5.4) | 1/12 (6.8) | 1/17 (11.6) | 1/14 (7.3) | 1/16 (9.2) |
| HIT-RATE@1 | 2/17 (10.9) | 1/13 (8.5) | 1/13 (6.1) | 1/12 (7.0) | 1/17 (10.4) | 1/12 (4.1) | 1/17 (5.8) |
| HIT-RATE@2 | 1/18 (10.8) | 2/15 (9.5) | 1/14 (5.9) | 1/12 (6.7) | 2/16 (11.0) | 1/13 (3.8) | 1/17 (6.0) |
| HIT-RATE@5 | 1/18 (10.3) | 1/15 (9.3) | 1/11 (5.8) | 1/14 (7.2) | 1/17 (11.3) | 1/14 (4.7) | 1/17 (7.6) |
| HIT-RATE@10 | 1/18 (10.0) | 1/14 (9.4) | 1/11 (6.0) | 1/15 (6.9) | 1/18 (11.4) | 1/14 (5.3) | 1/17 (8.1) |
| HIT-RATE@50 | 1/16 (9.7) | 1/15 (9.0) | 1/16 (5.4) | 1/12 (6.8) | 1/17 (11.6) | 1/14 (7.3) | 1/16 (9.2) |

Table 11: Continuation of Table 9 (Part 3 of 3).

| | I-neural | U-neural | Spectral-CF | DELF-MLP | DELF-EF | Mult-VAE | GlobalEffects | TopPop | Random |
|---|---|---|---|---|---|---|---|---|---|
| NDCG@1 | 13/17 (15.0) | 1/17 (10.2) | 19/23 (21.0) | 19/24 (21.0) | 10/18 (14.0) | 4/23 (13.0) | 2/19 (12.1) | 2/17 (10.0) | 10/23 (15.9) |
| NDCG@2 | 13/17 (15.6) | 4/19 (12.8) | 20/23 (21.6) | 20/24 (21.3) | 12/18 (14.3) | 7/20 (12.7) | 4/19 (13.4) | 3/18 (10.7) | 10/23 (16.4) |
| NDCG@5 | 12/18 (15.8) | 2/19 (13.0) | 20/24 (22.0) | 20/24 (21.7) | 7/18 (13.7) | 8/19 (12.1) | 4/19 (13.9) | 1/19 (11.1) | 10/23 (16.6) |
| NDCG@10 | 11/19 (15.4) | 3/18 (13.3) | 20/24 (22.4) | 20/24 (21.7) | 9/18 (14.4) | 3/19 (11.7) | 7/19 (14.0) | 1/18 (10.9) | 10/23 (16.7) |
| NDCG@50 | 13/19 (15.8) | 1/20 (13.4) | 21/24 (23.0) | 20/24 (21.9) | 12/19 (15.1) | 1/20 (10.4) | 7/19 (14.1) | 1/18 (10.1) | 9/22 (16.6) |
| PRECISION@1 | 13/17 (15.0) | 1/17 (10.2) | 19/23 (21.0) | 19/24 (21.0) | 10/18 (14.0) | 4/23 (13.0) | 2/19 (12.1) | 2/17 (10.0) | 10/23 (15.9) |
| PRECISION@2 | 10/17 (15.0) | 4/19 (13.0) | 20/23 (21.6) | 20/24 (21.3) | 12/18 (14.3) | 7/19 (12.4) | 4/19 (13.4) | 1/18 (10.6) | 10/23 (16.3) |
| PRECISION@5 | 12/19 (15.4) | 2/18 (13.1) | 20/24 (21.8) | 20/24 (21.7) | 7/18 (14.1) | 5/17 (11.4) | 2/19 (13.7) | 1/19 (10.9) | 10/22 (16.5) |
| PRECISION@10 | 12/19 (15.4) | 3/18 (13.3) | 20/24 (21.8) | 20/24 (21.4) | 11/17 (14.9) | 2/18 (10.8) | 7/19 (14.1) | 1/18 (10.7) | 10/22 (16.6) |
| PRECISION@50 | 13/19 (16.2) | 1/19 (12.9) | 22/24 (23.2) | 19/24 (21.6) | 13/19 (15.4) | 1/19 (8.6) | 7/20 (14.1) | 1/18 (9.3) | 9/22 (16.5) |
| RECALL@1 | 13/17 (15.0) | 1/17 (10.2) | 19/23 (21.0) | 19/24 (21.0) | 10/18 (14.0) | 4/23 (13.0) | 2/19 (12.1) | 2/17 (10.0) | 10/23 (15.9) |
| RECALL@2 | 10/17 (15.0) | 4/19 (13.0) | 20/23 (21.6) | 20/24 (21.3) | 12/18 (14.3) | 7/19 (12.4) | 4/19 (13.4) | 1/18 (10.6) | 10/23 (16.3) |
| RECALL@5 | 12/19 (15.4) | 2/18 (13.1) | 20/24 (21.8) | 20/24 (21.7) | 7/18 (14.1) | 5/17 (11.4) | 2/19 (13.7) | 1/19 (10.9) | 10/22 (16.5) |
| RECALL@10 | 12/19 (15.4) | 3/18 (13.3) | 20/24 (21.8) | 20/24 (21.4) | 11/17 (14.9) | 2/18 (10.8) | 7/19 (14.1) | 1/18 (10.7) | 10/22 (16.6) |
| RECALL@50 | 13/19 (16.2) | 1/19 (12.9) | 22/24 (23.2) | 19/24 (21.6) | 13/19 (15.4) | 1/19 (8.6) | 7/20 (14.1) | 1/18 (9.3) | 9/22 (16.5) |
| HIT-RATE@1 | 13/17 (15.0) | 1/17 (10.2) | 19/23 (21.0) | 19/24 (21.0) | 10/18 (14.0) | 4/23 (13.0) | 2/19 (12.1) | 2/17 (10.0) | 10/23 (15.9) |
| HIT-RATE@2 | 10/17 (15.0) | 4/19 (13.0) | 20/23 (21.6) | 20/24 (21.3) | 12/18 (14.3) | 7/19 (12.4) | 4/19 (13.4) | 1/18 (10.6) | 10/23 (16.3) |
| HIT-RATE@5 | 12/19 (15.4) | 2/18 (13.1) | 20/24 (21.8) | 20/24 (21.7) | 7/18 (14.1) | 5/17 (11.4) | 2/19 (13.7) | 1/19 (10.9) | 10/22 (16.5) |
| HIT-RATE@10 | 12/19 (15.4) | 3/18 (13.3) | 20/24 (21.8) | 20/24 (21.4) | 11/17 (14.9) | 2/18 (10.8) | 7/19 (14.1) | 1/18 (10.7) | 10/22 (16.6) |
| HIT-RATE@50 | 13/19 (16.2) | 1/19 (12.9) | 22/24 (23.2) | 19/24 (21.6) | 13/19 (15.4) | 1/19 (8.6) | 7/20 (14.1) | 1/18 (9.3) | 9/22 (16.5) |

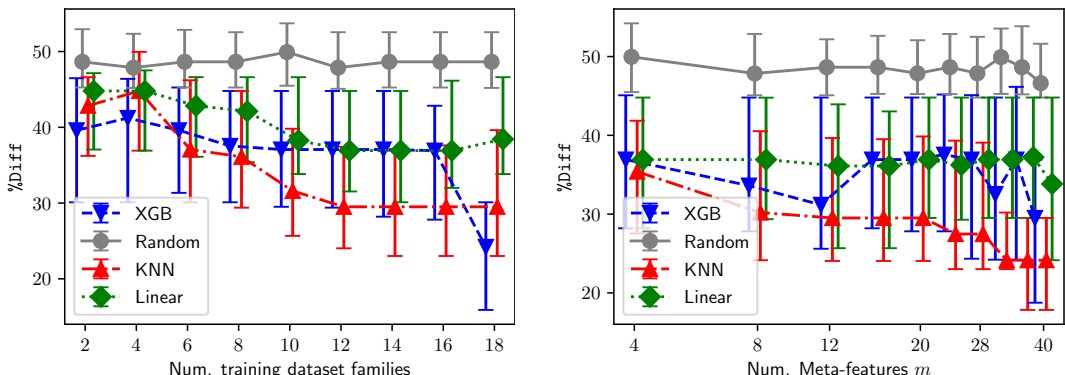

Figure 5: Performance of the RecZilla pipeline improves as we add more training meta-datapoints, and more meta-features $m$. Subsets of training meta-datapoints and meta-features are selected randomly, over 50 random trials. Points show the median %Diff, and error bars show the 40th and 60th percentile over all folds and random trials. (Left) $m = 10$ meta-features, while the number of training dataset families varies. (Right) All training data is used, while $m$ varies.

section of the README. In particular, any one of the 315 metrics can be used as the objective (or any computable function of these metrics). Once the meta-model is trained, then the above instructions can be used.

**Final takeaways.** Our work gives both intuition for working with rec-sys models, as well as a meta-learning pipeline that substantially lowers the level of human involvement when designing a high-performing rec-sys model. Specifically, practitioners can use our large-scale empirical study to gain insights such as how generalizable a given algorithm is, and how "hard" it will be to train a high-performing model, compared to other datasets. On the other hand, practitioners can follow the steps in our README to quickly train a high-performing model when faced with a brand new dataset.

## C.3 Additional experiments from Section 3

In this section, we give more experiments with Reczilla. Figure 5 (left) displays %Diff vs. the size of the meta-training set, and Figure 5 (right) displays the results of an ablation study on the number of selected meta-features $m$. All results are averaged over all leave-one-out folds and 50 random trials.

Generally, XGBoost and KNN outperform the linear model and the random baseline, with XGBoost achieving top performance when using 10 meta-features and the maximum number of training datasets. Furthermore, the number of datasets in the meta-training set matters more than the meta-learning model itself. For example, the improvement of XGBoost from 4 to 10 and to 18 training datasets is larger than the difference in performance between XGBoost and KNN at 4 and 10 training datasets, respectively. Finally, we find that the optimal number of meta-features for XGBoost and KNN peaks between 10 and 40.

## C.4 Meta-learner results with additional metrics

The results from Figures 4 and 5 were generated using PREC@10 as the base recommender system metric. Now, we re-run these experiments using COVERAGE@50 and HIT-RATE@5 as the base metrics, to show variety. See Figure 6.

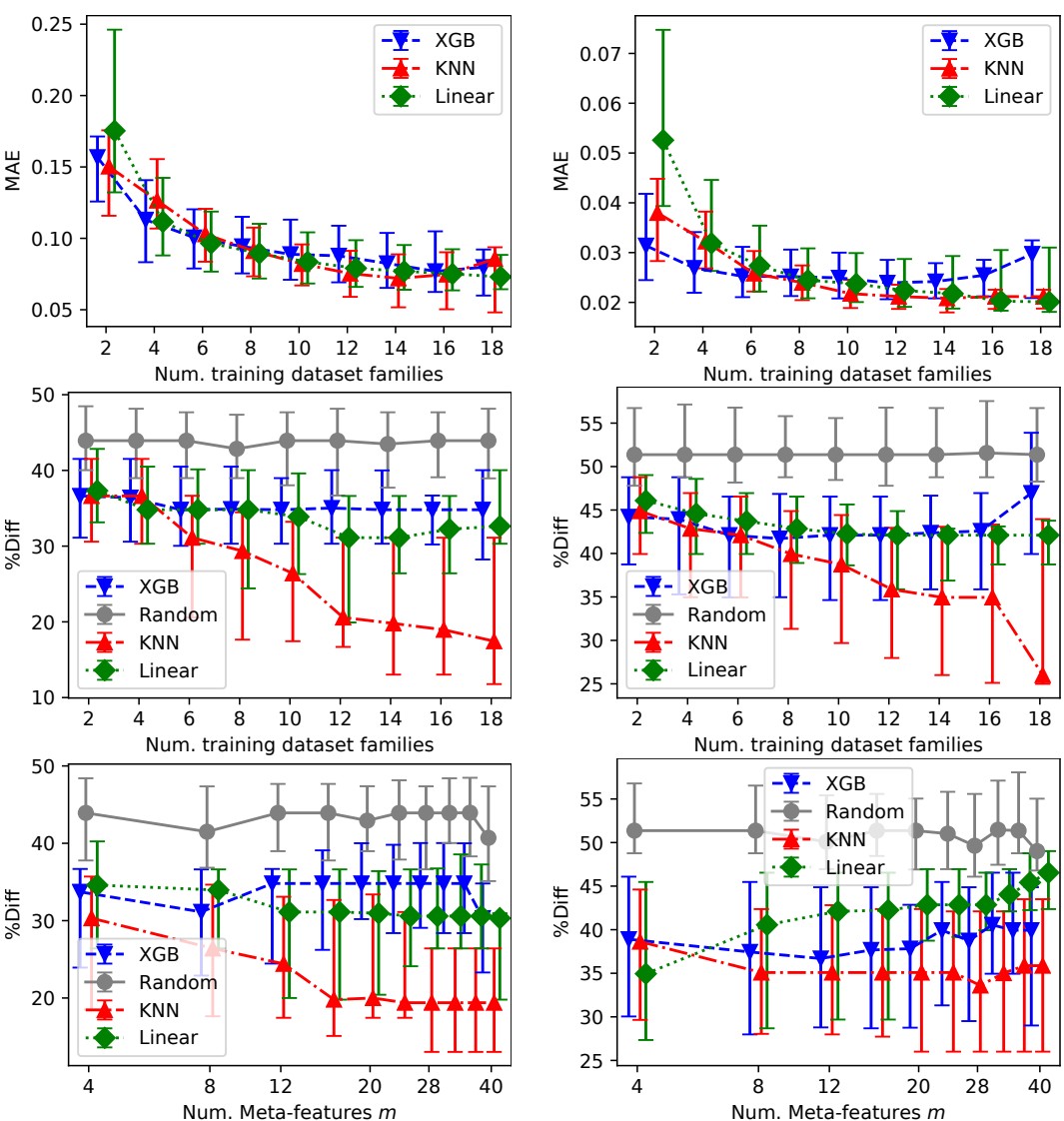

Figure 6: Performance of RecZilla when other base metrics are used: COVERAGE@50 (left column) and HIT-RATE@5 (right column). The first row shows MAE vs. num training dataset families (similar to Figure 4), The second row shows %Diff vs. num. training dataset families, and the third row shows %Diff vs. num. meta-features (similar to Figure 5). All results are averaged over all folds of leave-one-out validation and 50 trials.